# Protein phosphatase 1 in association with Bud14 inhibits mitotic exit in *Saccharomyces cerevisiae*

Dilara Kocakaplan[†], Hüseyin Karabürk[†], Cansu Dilege, Idil Kirdök, Seyma Nur Bektas, Ayse Koca Caydasi*

Department of Molecular Biology and Genetics, Koç University, Istanbul, Turkey

**Abstract** Mitotic exit in budding yeast is dependent on correct orientation of the mitotic spindle along the cell polarity axis. When accurate positioning of the spindle fails, a surveillance mechanism named the spindle position checkpoint (SPOC) prevents cells from exiting mitosis. Mutants with a defective SPOC become multinucleated and lose their genomic integrity. Yet, a comprehensive understanding of the SPOC mechanism is missing. In this study, we identified the type 1 protein phosphatase, Glc7, in association with its regulatory protein Bud14 as a novel checkpoint component. We further showed that Glc7-Bud14 promotes dephosphorylation of the SPOC effector protein Bfa1. Our results suggest a model in which two mechanisms act in parallel for a robust checkpoint response: first, the SPOC kinase Kin4 isolates Bfa1 away from the inhibitory kinase Cdc5, and second, Glc7-Bud14 dephosphorylates Bfa1 to fully activate the checkpoint effector.

**\*For correspondence:**
aykoca@ku.edu.tr

[†]These authors contributed equally to this work

**Competing interest:** The authors declare that no competing interests exist.

## Introduction

Budding yeast undergoes asymmetric cell division in every cell cycle and is an intrinsically polarized cell. In every cell cycle, a new cell buds on the mother cell and pinches of from this location at the end of the cell division, giving rise to a young daughter cell and an old mother cell. In order for the daughter cell to receive one copy of the duplicated genetic material, budding yeast has to segregate its chromosomes along its mother-to-bud polarity axis, which requires positioning of the mitotic spindle apparatus along this direction. The correct orientation of the mitotic spindle is achieved by two redundant pathways, namely the 'early' and 'late' pathways, which are dependent on the adenomatous polyposis coli (APC) homolog Kar9 and the microtubule motor protein Dyn1, respectively (*Adames and Cooper, 2000*; *Beach et al., 2000*; *Li et al., 1993*; *Siller and Doe, 2009*).

Correct orientation of the mitotic spindle is monitored by a mitotic checkpoint named the spindle position checkpoint (SPOC) in budding yeast. SPOC prevents cell cycle progression of cells that fail to orient their spindle in the mother-to-bud direction, and hence provides time for cells to correct their spindle orientation before mitotic exit (*Adames et al., 2001*; *Bardin et al., 2000*; *Bloecher et al., 2000*; *Pereira et al., 2000*; *Yeh et al., 1995*). Yeast cells with defects in the spindle positioning pathways or cells with impaired microtubule function rely on the SPOC to maintain their ploidy. Absence of SPOC causes multinucleation, enucleation, and aneuploidy in budding yeast. A SPOC-like checkpoint also exists in *Drosophila* and may be present in higher eukaryotes (*Cheng et al., 2008*; *O'Connell and Wang, 2000*; *Pereira and Yamashita, 2011*).

In response to spindle mispositioning, SPOC inhibits cell cycle progression by inhibiting the mitotic exit network (MEN) (*Baro et al., 2017*; *Caydasi et al., 2010a*; *Caydasi and Pereira, 2012*; *Weiss, 2012*). MEN is a GTPase-driven signaling pathway that is essential for mitotic exit in budding yeast (*Jaspersen et al., 1998*; *Shou et al., 1999*). A single conserved phosphatase, named Cdc14, triggers mitotic exit in budding yeast through inactivation of the mitotic cyclin-dependent kinase (CDK) and dephosphorylation of CDK targets (*Manzano-López and Monje-Casas, 2020*). At the anaphase

onset, a transient wave of Cdc14 activation is triggered by the cdc fourteen early anaphase release (FEAR) network, whereas later in anaphase the MEN promotes full activation of Cdc14 (*Jaspersen et al., 1998*; *Lee et al., 2001a*; *Rock and Amon, 2009*; *Shou et al., 1999*). A Ras-like small GTPase named Tem1 is the main switch located near the top of the MEN (*Lee et al., 2001a*; *Morishita et al., 1995*; *Scarfone and Piatti, 2015*; *Shirayama et al., 1994b*). When bound to GTP, Tem1 initiates the MEN signaling by recruiting the Cdc15 kinase to the spindle pole bodies (SPBs, centrosome equivalent in yeast), which therein activates Dbf2 kinase in association with its regulatory subunit Mob1 (*Asakawa et al., 2001*; *Cenamor et al., 1999*; *Visintin and Amon, 2001*).

SPOC inhibits the MEN through inactivation of Tem1 by a bipartite GTPase-activating complex (GAP) composed of Bfa1 and Bub2 proteins (*Geymonat et al., 2002*). Activity of the Bfa1-Bub2 GAP complex is tightly regulated at the level of Bfa1 phosphorylation. In cells with a correctly positioned anaphase spindle, the polo-like kinase Cdc5 phosphorylates Bfa1 to prevent Bfa1-Bub2 GAP activity towards Tem1 (*Hu et al., 2001*). CDK also phosphorylates Bfa1 in an activating manner until anaphase when this phosphorylation is removed in a FEAR-related manner (*Caydasi et al., 2017*). In the presence of spindle misalignment, however, Kin4 kinase phosphorylates Bfa1 (*Maekawa et al., 2007*). This phosphorylation prevents inhibitory phosphorylation of Bfa1 by Cdc5 kinase (*Pereira and Schiebel, 2005*). Consequently, Bfa1-Bub2 GAP complex activity is promoted, Tem1 activity is prevented, and exit from mitosis is inhibited.

SPBs, serving as a scaffold, play a key role in mitotic exit and its timely regulation by the spindle position (*Campbell et al., 2020*; *Gruneberg et al., 2000*; *Pereira and Schiebel, 2001*; *Valerio-Santiago and Monje-Casas, 2011*). When the spindle is properly oriented, Bfa1-Bub2 and Tem1 reside in a preferential, asymmetrical manner at the SPB that moves to the daughter cell (dSPB) (*Molk et al., 2004*; *Pereira et al., 2000*). On the contrary, Kin4 localizes exclusively to the SPB that stays in the mother cell (mSPB) during an unperturbed anaphase (*D'Aquino et al., 2005*; *Pereira and Schiebel, 2005*). Spindle misalignment alters SPB localization of SPOC components. Upon mispositioning of the mitotic spindle, Kin4 localizes to both SPBs (*Pereira and Schiebel, 2005*). Phosphorylation of Bfa1 by Kin4 promotes binding of the 14-3-3 family protein Bmh1 to Bfa1, which causes dissociation of Bfa1-Bub2 from SPBs (*Caydasi et al., 2012*; *Caydasi et al., 2014*; *Caydasi and Pereira, 2009*; *Monje-Casas and Amon, 2009*). Hence, upon spindle mispositioning, the amount of Bfa1-Bub2 at both SPBs decreases, while their cytoplasmic levels increase. The release of Bfa1-Bub2 from SPBs is thought to keep Bfa1-Bub2 away from the inhibitory kinase Cdc5 and thus prevent inactivation of Bfa1-Bub2 by Cdc5. However, what dephosphorylates the Cdc5-phosphorylated Bfa1 remains elusive.

Here, we identified Bud14 as a novel SPOC protein. Deletion of *BUD14* rescued cold sensitivity of *lte1Δ* cells, lethality of mitotic exit-deficient *lte1Δ spo12Δ* cells, and growth defects of MEN temperature-sensitive mutants *cdc15-1*, *dbf2-2*, and *mob1-67*. Our data showed that *bud14Δ* cells accumulated multinucleated phenotypes when spindle positioning was impaired in cells via deletion of *KAR9* or *DYN1*. Fluorescence time-lapse microscopy revealed that *bud14Δ* cells failed to delay mitotic exit in response to spindle mispositioning. We observed an additive decrease in anaphase duration of *kin4Δ bud14Δ* cells with misaligned spindles compared to the cells bearing single-gene deletions of *KIN4* or *BUD14,* suggesting that Kin4 and Bud14 work in parallel in SPOC. We further found that the mitotic exit inhibitory function Bud14 required its association with the type 1 protein phosphatase, Glc7. A temperature-sensitive version of Glc7 (*glc7-12*) and a version of Bud14 that cannot interact with Glc7 (*bud14-F379A*) caused SPOC deficiency and rescued growth of mitotic exit mutants. Yeast two hybrid data indicated an interaction between Bfa1 and Bud14 that required the presence of Bub2, which suggests that Bud14 may recognize Bfa1-Bub2. Intriguingly, *bud14Δ* and *bud14-F379A* cells had more Bfa1-Bub2 localized at the dSPBs, suggestive of a role for Bud14-Glc7 in limiting SPB-bound levels of Bfa1-Bub2. Similarly to the wild-type cells, upon spindle mispositioning, levels of Bfa1 at the dSPB were able to decrease in *bud14Δ* cells, unlike *kin4Δ* cells, which also supported separate functions of Bud14 and Kin4 in SPOC. We further observed that *lte1Δ* cells lacking Bud14 accumulated hyperphosphorylated Bfa1 forms during anaphase. Furthermore, overexpression of *BUD14* but not *bud14-F379A* caused a reduction in this hyperphosphorylated forms of Bfa1. Finally, through in vitro phosphatase assays we showed that Glc7-Bud14 promoted dephosphorylation of hyperphosphorylated Bfa1. Our data altogether indicates that Glc7-Bud14 is a novel mitotic exit inhibitor in budding yeast that works upon spindle mispositioning. We propose a checkpoint model in which two independent mechanisms activate the checkpoint effector: first, the SPOC kinase Kin4 prevents further

Cdc5 phosphorylation of Bfa1 by rapidly removing Bfa1 away from the Cdc5 kinase, and second, Glc7-Bud14 dephosphorylates Bfa1 to activate the Bfa1-Bub2 GAP complex. Both mechanisms are crucial for rapid activation of the checkpoint effector Bfa1-Bub2 upon mispositioning of the anaphase spindle.

## Results

### Bud14 is a novel mitotic exit inhibitor

Lte1 is a mitotic exit activator that becomes essential for mitotic exit at cold temperatures (<16°C) (*Shirayama et al., 1994a*). Although the exact function of Lte1 in mitotic exit is not fully understood, one of the ways that Lte1 promotes mitotic exit is by preventing binding of the SPOC kinase Kin4 to the SPB that has migrated into the bud and by inhibiting its activity therein (*Bertazzi et al., 2011*; *Falk et al., 2011*). At physiological temperatures (i.e., 30°C), *lte1Δ* cells rely on the presence of the FEAR network to exit mitosis. Deletion of FEAR network components such as *SPO12* or *SLK19* in *lte1Δ* cells causes lethality due to failure of mitotic exit (*Stegmeier et al., 2002*). Lethality of the *lte1Δ spo12Δ* double mutants can be rescued by deletion of the known mitotic exit inhibitors *BFA1*, *BUB2*, and *KIN4* (*D'Aquino et al., 2005*; *Stegmeier et al., 2002*). In order to identify novel mitotic exit inhibitors, we designed a genetic screen that looked for single-gene deletions rescuing *lte1Δ spo12Δ* lethality. In addition to the known mitotic exit inhibitors *BFA1*, *BUB2*, and *KIN4*, we identified *BUD14* as a novel gene that contributes to the mitotic arrest of *lte1Δ spo12Δ* cells (data not shown). Deletion of *BUD14* rescued lethality of *lte1Δ spo12Δ* cells (*Figure 1A*). Deletion of *BUD14* also rescued the cold sensitivity of *lte1Δ* cells (*Figure 1B*). This data indicates that Bud14 is a novel inhibitor of mitotic exit.

We next asked whether deletion of *BUD14* rescues growth lethality of MEN mutants. For this, we deleted *BUD14* in several MEN temperature-sensitive (ts) mutants (*Jaspersen et al., 1998*) and compared their growth at different temperatures. To allow for better comparison, we also deleted *BFA1*, a known mitotic exit inhibitor, in the same MEN-ts mutants (*Caydasi et al., 2017*; *Hu et al., 2001*; *Pereira et al., 2000*; *Scarfone et al., 2015*; *Wang et al., 2000*). In line with its mitotic exit inhibitory role, deletion of *BFA1* rescued the lethality of *mob1-67, dbf2-2, cdc15-1,* and *cdc5-10* ts mutants at 33, 37, 35, and 37°C, respectively (*Figure 1C*). Deletion of *BUD14* also promoted the growth of *mob1-67, dbf2-2,* and *cdc15-1* mutants at 33, 35, and 35°C, respectively (*Figure 1C*). In all cases, growth rescue by *BUD14* deletion was milder than that of *BFA1* deletion. Furthermore, *BUD14* deletion did not rescue the growth of the *cdc5-10* while *BFA1* deletion did (*Figure 1C*), suggesting that the mitotic exit inhibitory function of Bud14 does not contribute to *cdc5-10* lethality. Deletion of neither *BFA1* nor *BUD14* rescued the growth of the *cdc14-2* mutant (*Figure 1C*). Cdc14 is the essential phosphatase that is activated at the very downstream of the MEN pathway. The fact that *bud14Δ* and *bfa1Δ* does not rescue *cdc14-2* supports their function upstream of the MEN pathway.

### Bud14 is essential for SPOC functionality

The known mitotic exit inhibitors Bfa1, Bub2, and Kin4 are key components of the SPOC, which is a surveillance mechanism that inhibits the MEN when anaphase spindle fails to orient in the mother-to-daughter direction. We thus asked whether the novel mitotic exit inhibitor Bud14 has a role in SPOC. To address this, we first employed a population-based approach. Cells with a functional SPOC, arrest in anaphase with mispositioned nuclei upon spindle mispositioning, whereas SPOC-deficient cells such as *bfa1Δ* or *kin4Δ* continue cell cycle progression, resulting in the formation of multinucleated and enucleated cells. We assayed SPOC deficiency in cell populations as a measure of frequency of multinucleation, relative to that of the spindle mispositioning (SPOC deficiency index). Kar9- and Dynein-dependent spindle positioning pathways act redundantly to align the spindle along mother-to-daughter polarity axis in budding yeast (*Eshel et al., 1993*; *Li et al., 1993*; *Miller and Rose, 1998*). To induce spindle mispositioning, we first made use of *kar9Δ* cells. As expected, deletion of known SPOC components *KIN4* or *BFA1* in the *kar9Δ* background yielded a remarkably high degree of SPOC deficiency index (*Figure 2A*). Notably, deletion of *BUD14* in the *kar9Δ* background also caused a significantly high SPOC deficiency index (*Figure 2A*). Similar results were obtained using a *dyn1Δ* background (*Figure 2B*). These data suggest that *bud14Δ* cells are defective in SPOC.

We then employed a single-cell analysis approach to assay SPOC functionality. For this, we analyzed the timing of mitotic exit in *bud14Δ* cells during spindle misalignment and normal alignment in a

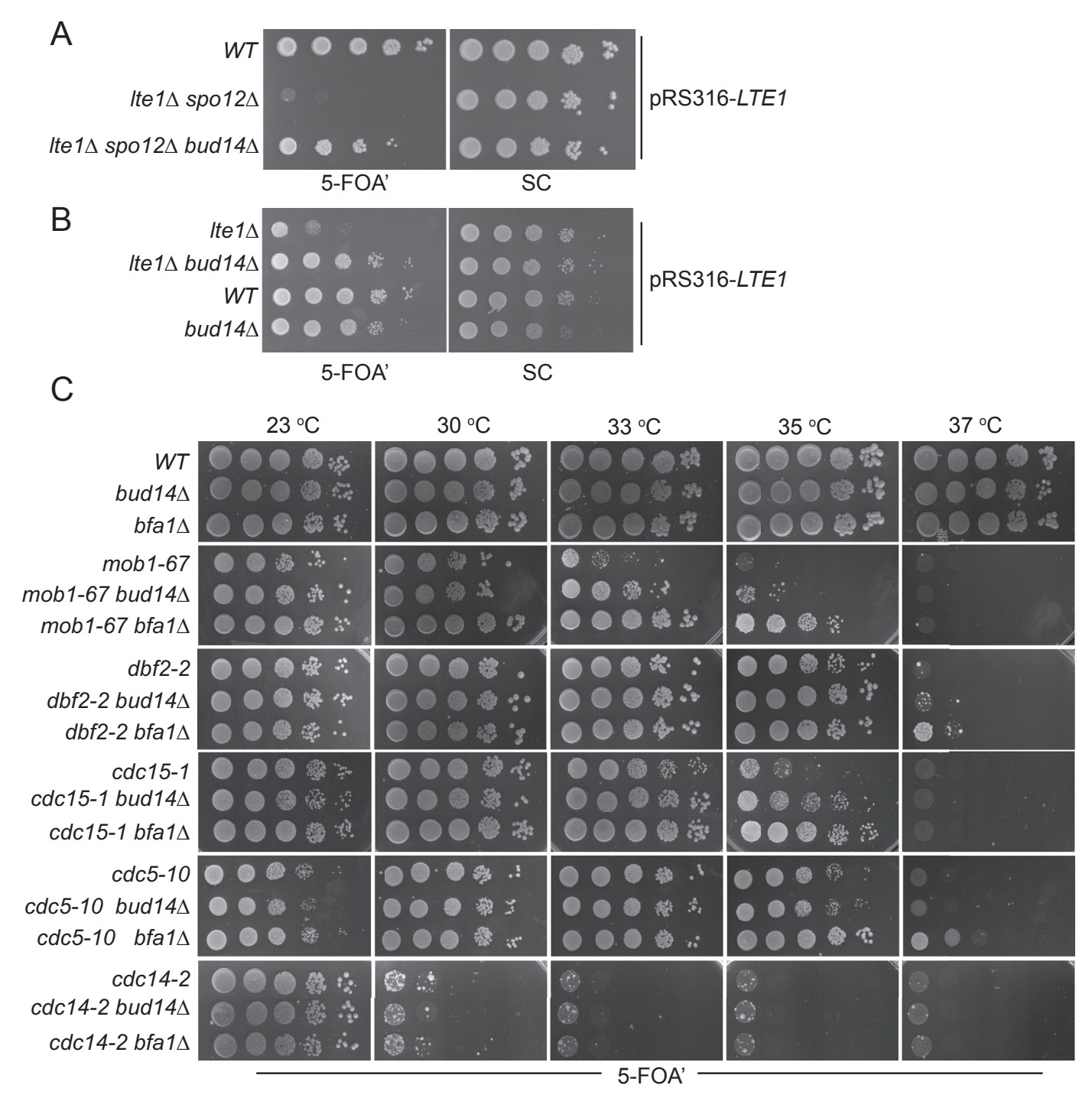

**Figure 1.** Bud14 deletion rescues growth of cells with impaired mitotic exit. (**A**) *bud14Δ* cells rescue the synthetic lethality of *lte1Δ spo12Δ* cells. (**B**) *bud14Δ* cells rescue the lethality of *lte1Δ* cells at 18°C. (**C**) Comparison of growth rescue of mitotic exit network temperature-sensitive (MEN-ts) mutants upon deletion of *BUD14* and *BFA1*. Serial dilutions of indicated strains were spotted on indicated plates and grown at given temperatures. 5-Fluoroorotic acid (5-FOA) plates negatively select for the *URA3*-based plasmids (pRS316 containing the *LTE1* in **A** and **B**, pRS316 containing the wild-type gene copy of the corresponding MEN mutant in **C**). Thus, only cells that have lost these plasmids can grow on 5-FOA plates where genetic interactions can be observed.

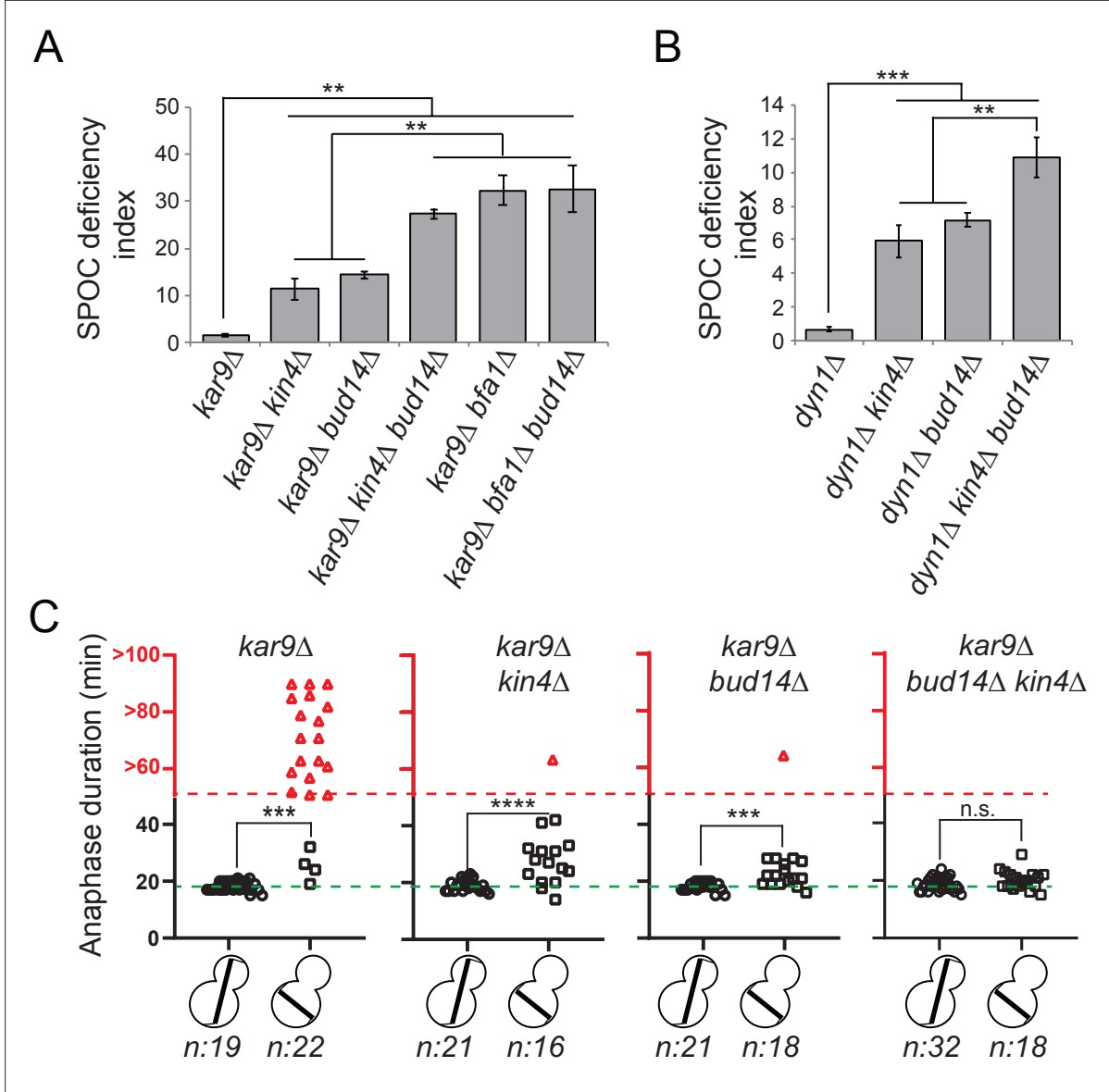

**Figure 2.** *bud14Δ* cells are spindle position checkpoint (SPOC) deficient. (**A, B**) Endpoint analysis of SPOC integrity of indicated yeast strains in *kar9Δ* (**A**) and *dyn1Δ* (**B**) background. Cells were fixed with ethanol and stained with DAPI. Cells with normally aligned nuclei, mispositioned nuclei, and multinucleated cells that failed to correctly position their spindle before mitotic exit were counted by microscopy and their SPOC deficiency index percentages were plotted, where SPOC deficiency index = % cells with multinucleation ÷ % cells with mispositioned nuclei × 10. Graphs are average of three independent experiments. A minimum of 100 cells were counted from each strain in each experiment. Error bars show standard deviation. Two-tailed Student's t-test was applied. **p<0.01, ***p<0.001. (**C**) Single-cell analysis of SPOC integrity in indicated strains. Duration of anaphase in cells with misaligned and normally aligned spindles was calculated as explained in the text and plotted as dotplots. Data points in red indicate the cells in which the spindle did not break down during the time-lapse movie. In this case, plotted values are the time duration during which these cells were observed in anaphase through the time-lapse movie. Consequently, the actual anaphase duration is greater than the value plotted in red. This fact is emphasized in the red part of the y-axis, indicated above the red dashed line, with addition of the '>' symbol before the y-axis values. Green dash line indicates the mean value of the anaphase duration in cells with normally aligned spindles. One-way ANOVA with uncorrected Fisher's LSD was applied for statistical analysis. ***p<0.001, ****p<0.0001. n: sample size. All pairwise comparisons and descriptive statistics are shown in the corresponding source data files, whereas only comparisons of normal and misaligned spindles are shown in the figure.

The online version of this article includes the following source data and figure supplement(s) for figure 2:

**Source data 1.** Numerical data and statistics for *Figure 2A*.

**Source data 2.** Numerical data and statistics for *Figure 2B*.

**Source data 3.** Numerical data and statistics for *Figure 2C*.

*Figure 2 continued on next page*

*Figure 2 continued*

**Figure supplement 1.** Bud14 does not influence Kin4 function.

**Figure supplement 1—source data 1.** Numerical data for *Figure 2—figure supplement 1B*.

**Figure supplement 1—source data 2.** Numerical data for *Figure 2—figure supplement 1C*.

**Figure supplement 1—source data 3.** Labeled uncropped blots images for *Figure 2—figure supplement 1D*.

**Figure supplement 1—source data 4.** Raw scans of the x-ray films for (**A**) anti-HA blot, (**B**) anti-Clb2 blot, and (**C**) anti-tubulin blot.

*GFP-TUB1 kar9Δ* cell background. *GFP-TUB1* served as a spindle marker. *kar9Δ* population contains cells with misaligned spindles as well as cells with normally aligned spindles, and thus allows analysis of anaphase duration in both cases. Cells were imaged through fluorescence time-lapse microscopy for 90 min with 1 min time resolution at 30°C. Anaphase duration was calculated as the time elapsed between the start of fast spindle elongation and the spindle breakage, which are indications of anaphase onset and mitotic exit, respectively. In cells with a normally positioned spindle, mean anaphase duration of all analyzed cell types was similar – ranging from 18 to 19 min with a standard deviation of 2 min (*Figure 2C*, *Figure 2—figure supplement 1—source data 2a*). The majority of *kar9Δ* cells with a misaligned spindle stayed arrested with an intact spindle during the time-lapse movie (*Figure 2C*, *kar9Δ*, red triangles), whereas *kin4Δ kar9Δ* cells with misaligned spindles broke their anaphase spindle in average 27 ± 7 min (mean ± standard deviation) after the onset of anaphase (*Figure 2C*, *kar9Δ kin4Δ*, black squares). *bud14Δ kar9Δ* cells with misaligned spindles also broke their spindle and with a mean anaphase duration of 23 ± 4 min (mean ± standard deviation; *Figure 2C*, *kar9Δ bud14Δ*, black squares). These data altogether suggest that Bud14 is needed as a part of the SPOC mechanism to delay mitotic exit upon spindle mispositioning.

## Function of Bud14 in SPOC is parallel to Kin4

Bfa1-Bub2 is the most downstream component of SPOC, and Kin4 is the key SPOC kinase that resides upstream of Bfa1 to promote Bfa1-Bub2 GAP activity upon spindle misalignment. We next asked whether Bud14 works parallel to Kin4 in SPOC. We reasoned that if Bud14 and Kin4 work in parallel, concomitant loss of Bud14 and Kin4 would cause a greater SPOC deficiency index than individual loss of Bud14 or Kin4. SPOC deficiency index of *kin4Δ bud14Δ kar9Δ* was significantly higher than that of *kin4Δ kar9Δ* or *bud14Δ kar9Δ* (*Figure 2A*). Likewise, SPOC deficiency phenotype was more prominent in *kin4Δ bud14Δ dyn1Δ* cells than in *kin4Δ dyn1Δ* or *bud14Δ dyn1Δ* cells (*Figure 2B*). Thus, *KIN4* and *BUD14* deletion showed an additive effect with respect to SPOC deficiency. Of note, deletion of *BUD14* in *bfa1Δ kar9Δ* cells did not cause any increase in SPOC deficiency index compared to *bfa1Δ kar9Δ* cells (*Figure 2A*). These data are suggestive of a role for Bud14 parallel to Kin4 and upstream of Bfa1 and in SPOC.

Additive effect of *KIN4* and *BUD14* deletion was also evident in anaphase duration comparisons. Even though *BUD14* deletion caused mitotic exit of cells with misaligned spindles, duration of anaphase in *bud14Δ kar9Δ* cells was significantly longer during spindle misalignment than during spindle normal alignment (*Figure 2C*, *bud14Δ*, squares vs. circles). In addition, 1 out of 18 cells was able to hold the SPOC arrest longer than an hour. Similar phenomenon was also observed in *kar9Δ kin4Δ* cells (*Figure 2C*, *kin4Δ*, squares vs. circles) and was also reported before (*Caydasi et al., 2017*; *Falk et al., 2016b*). This indicates that in the absence of Bud14 or Kin4 cells are still able to delay mitotic exit in response to spindle positioning defects, albeit for a short time. On the contrary, when both *KIN4* and *BUD14* were deleted, cells exited mitosis with the same timing regardless of the spindle position (*Figure 2C*, *kar9Δ bud14Δ kin4Δ*, black squares vs. circles). Thus, concomitant deletion of *KIN4* and *BUD14* results in a more severe loss of SPOC functionality than single deletion of either *KIN4* or *BUD14*. This data is in line with our conclusion that Kin4 and Bud14 act in parallel to promote SPOC arrest.

To rule out the possibility that Bud14 may also be acting upon Kin4, we next analyzed the effect of Bud14 on Kin4 functionality. For this, we first analyzed cells overexpressing Kin4. Kin4 overexpression blocks exit from mitosis through constitutive Bfa1-Bub2 activation and thus results in lethality (*D'Aquino et al., 2005*). This lethality can be rescued by deletion of *BFA1* or *BUB2*, which are the most downstream SPOC components, and by deletion of *RTS1* or *ELM1*, which are Kin4 regulators, and by deletion of *BMH1*, which works downstream of *KIN4* (*Caydasi et al., 2010b*; *Caydasi et al., 2014*).

Unlike the aforementioned gene deletions, deletion of *BUD14* did not rescue the lethality of Kin4 overexpression (*Figure 2—figure supplement 1A*). Thus, overproduced Kin4 is still able to activate Bfa1 constitutively in the absence of Bud14. Next, we analyzed the effect of Bud14 on Kin4 localization. Kin4 localized to SPBs in response to microtubule depolymerization independently of the presence of Bud14 (*Figure 2—figure supplement 1B*). Likewise, Kin4 localized to SPBs during anaphase in *lte1Δ* cells both in the presence and absence of Bud14 (*Figure 2—figure supplement 1C*). Kin4 mobility on SDS-PAGE was not altered in cells lacking Bud14 either (*Figure 2—figure supplement 1D*). These data indicate that Bud14 does not work upstream of Kin4. As we will describe later, analysis of Bfa1 localization during spindle misalignment also supported that Kin4 regulation of Bfa1 takes place independently of Bud14 (Figure 6D, *Figure 6—figure supplement 3*). Taken altogether, our data support a model where Bud14 acts in a pathway parallel to Kin4 in SPOC.

## Bud14 function in SPOC is independently of its role in actin regulation

Bud14 regulates actin cable formation through its formin-regulatory motif (amino acids 135–150) (*Chesarone et al., 2009*; *Eskin et al., 2016*). Bud14's function in controlling formin activity also requires its interaction with KELCH domain proteins Kel1 and Kel2 (*Gould et al., 2014*). We addressed the contribution of actin regulatory role of Bud14 to its mitotic exit inhibitory function through two independent approaches. First, we made use of the *bud14-5A* mutant whose formin-regulatory motif is disrupted by five alanine amino acid substitutions at Bud14 residues 135 and 137–140 (*Eskin et al., 2016*). Ectopic expression of *bud14-5A* and *BUD14* on a centromeric yeast plasmid (pRS416) under *BUD14* endogenous/native promoter rescued the SPOC-deficient phenotype of *bud14Δ kar9Δ* cells (*Figure 3A*; *Figure 3—figure supplement 1A*). Similarly, ectopic expression of *bud14-5A* in *lte1Δ bud14Δ* cells complemented *BUD14* deletion by restoring cold sensitivity of *lte1Δ* (*Figure 3B*). Thus, *bud14-5A* mutant, which is deficient in formin binding, is functional in terms of mitotic exit inhibition. Of note, expression of *bud14-5A* caused a significantly greater extent of reduction in SPOC deficiency index of *bud14Δ kar9Δ* cells than expression of *BUD14* (*Figure 3A*). Furthermore, Bud14-5A interacted with Glc7 more readily in yeast two hybrid assays (*Figure 3—figure supplement 1B*), which may indicate a link between Glc7-Bud14 interaction and SPOC robustness.

Second, we made use of *kel1Δ* and *kel2Δ* cells. Kel1, Kel2, and Bud14 form a stable complex that regulates the formin Bnr1 and any missing protein from this complex causes actin cable defects to the same extent (*Gould et al., 2014*). We reasoned that if Kel1-Kel2-Bud14 complex, which is crucial for actin regulation, is also important for SPOC functionality, deletion of *KEL1* or *KEL2* to disturb the complex would also cause SPOC deficiency similar to the *bud14Δ* cells. However, this was not the case. Unlike *bud14Δ kar9Δ* cells, *kel1Δ kar9Δ* or *kel2Δ kar9Δ* cells did not display SPOC deficiency (*Figure 3C*). These data altogether suggest that Bud14 function in SPOC is independent of its role in actin cable regulation.

Interestingly, deletion of *KEL1* also rescues cold sensitivity of *lte1Δ* similar to the deletion of *BUD14* (*Höfken and Schiebel, 2002*; *Seshan et al., 2002*; *Figure 3D*). Synthetic lethality of *lte1Δ spo12Δ*, on the other hand, can only be slightly rescued by the deletion of *KEL1*, whereas deletion of *BUD14* in *lte1Δ spo12Δ* cells causes a more pronounced growth rescue phenotype (*Figure 3E*). Deletion of *KEL2,* however, does not rescue *lte1Δ spo12Δ* lethality, while it only has a minor impact on cold sensitivity of *lte1Δ* cells (*Figure 3D and E*). These data agree with previous reports that Kel1 and Kel2 have a mitotic exit inhibitory function (*Geymonat et al., 2010*; *Höfken and Schiebel, 2002*; *Seshan et al., 2002*) and suggest that this function of Kel1 and Kel2 in mitotic exit inhibition differs from that of Bud14 as Kel1/Kel2 are not influencing the SPOC.

## Bud14 inhibits mitotic exit through its interaction with Glc7

Bud14 is a regulatory subunit of the type I protein phosphatase (PP1) Glc7 (*Cullen and Sprague, 2002*; *Knaus et al., 2005*; *Lenssen et al., 2005*). We next asked whether Bud14 exerts its mitotic exit inhibitory role through Glc7. To address this question, we used mutants with impaired Glc7-Bud14 function. First, we used a mutant version of Bud14 that cannot bind Gcl7 (*bud14-F379A*) (*Knaus et al., 2005*; *Figure 3—figure supplement 1*). Introduction of the wild-type *BUD14* but not the *bud14-F379A* mutant on a centromeric plasmid restored cold sensitivity of the *lte1Δ bud14Δ* (*Figure 4A*). Likewise, expression of *BUD14* but not *bud14-F379A* on a centromeric plasmid under *BUD14* endogenous/

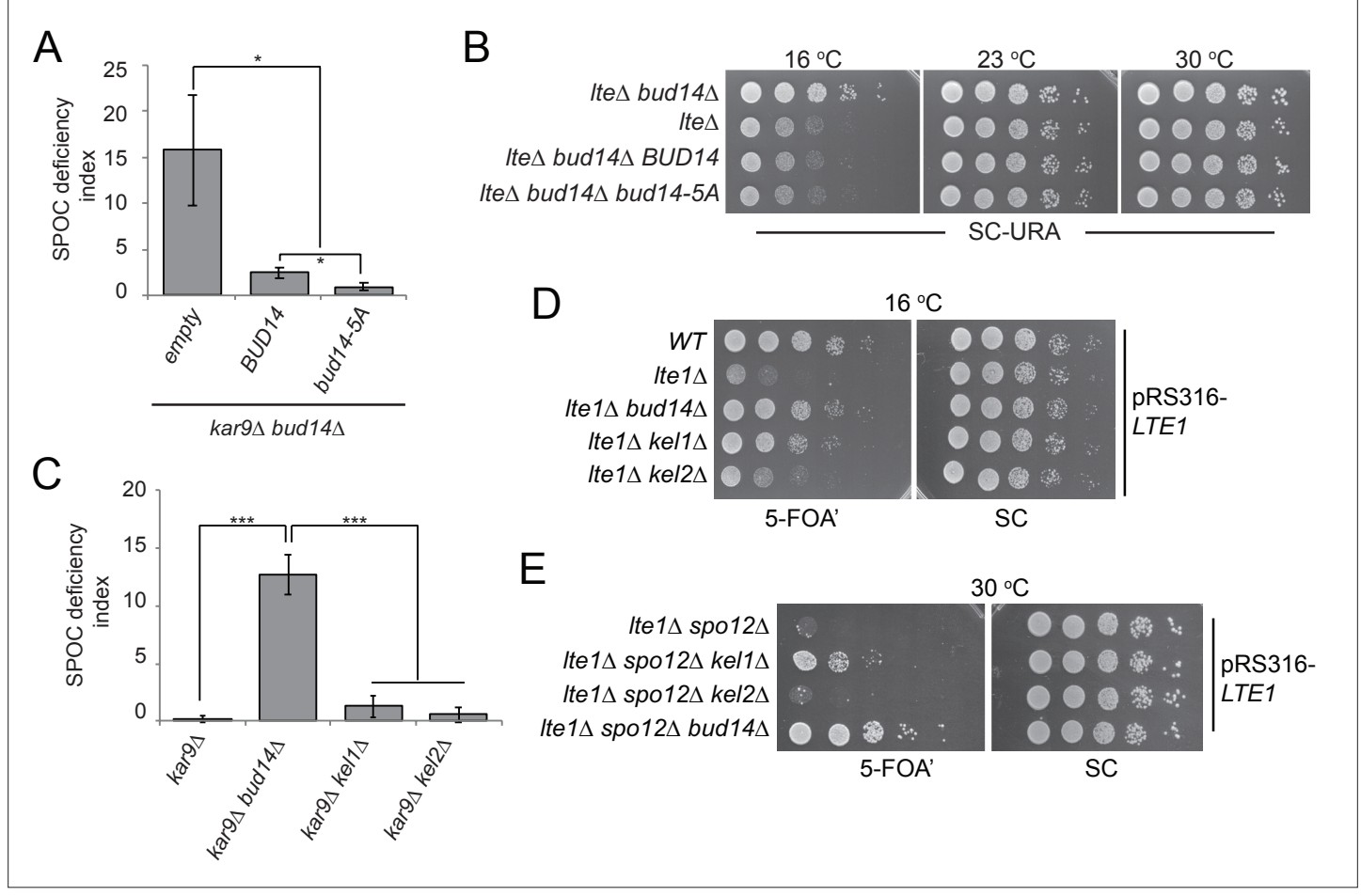

**Figure 3.** Function of Bud14 in actin regulation is dispensable for spindle position checkpoint (SPOC). (**A**) Endpoint analysis of SPOC deficiency index in *bud14Δ kar9Δ* cells carrying *URA3*-based empty plasmid (empty) or *BUD14*-containing *URA3*-based plasmids (*BUD14* and *bud14-5A*). Graphs are average of three independent experiments. A minimum of 100 cells were counted from each strain in each experiment. Error bars show standard deviation. *p<0.05 according to Student's t-test. (**B**) Serial dilutions of indicated strains bearing *URA3*-based empty plasmid (not indicated on figure) or *BUD14*-containing *URA3*-based plasmids (*BUD14* and *bud14-5A*) were spotted on SC-URA plate and grown at indicated temperatures. (**C**) Endpoint analysis of SPOC deficiency index in indicated cell types. Graphs are average of three independent experiments. A minimum of 100 cells were counted from each strain in each experiment. Error bars show standard deviation. ***p<0.001 according to Student's t-test. (**D, E**) Serial dilutions of indicated strains bearing *LTE1* on *URA3*-based pRS316 plasmid were spotted on 5-fluoroorotic acid (5-FOA) and SC plates and grown at indicated temperatures. 5-FOA negatively selects for *URA3*-containing plasmids, thus cells lose their pRS316-*LTE1* plasmids on 5-FOA plates and genetic interactions can be observed on this plate.

The online version of this article includes the following figure supplement(s) for figure 3:

**Source data 1.** Numerical data and statistics for *Figure 3A*.

**Source data 2.** Numerical data and statistics for *Figure 3C*.

**Figure supplement 1.** Expression and Glc7 binding of Bud14 mutants.

**Figure supplement 1—source data 1.** Labeled uncropped blot images for *Figure 3—figure supplement 1A*.

**Figure supplement 1—source data 2.** Raw scans of the blot images for (**A**) *Figure 3—figure supplement 1—source data 1*, anti-GFP blot, upper panel, and (**B**) *Figure 3—figure supplement 1—source data 1*, anti-tublin blot, lower panel.

native promoter rescued the SPOC-deficient phenotype of *bud14Δ kar9Δ* cells (*Figure 4B*). These data suggest that Glc7-Bud14 interaction is essential for Bud14 function in SPOC.

Next, we used the temperature-sensitive *glc7-12* mutant (*Andrews and Stark, 2000*; *Cheng and Chen, 2010*) to further confirm the contribution of Glc7 to mitotic exit inhibition. *glc7-12* exhibits a growth defect at 33–35°C and is lethal at 37°C, as an indication of Glc7 partial and full inactivation at these temperatures, respectively (*Figure 4—figure supplement 1A*, YPD plates; *Andrews and Stark, 2000*; *Cheng and Chen, 2010*). Partial inactivation of *glc7-12* at 33–35°C is also evident when

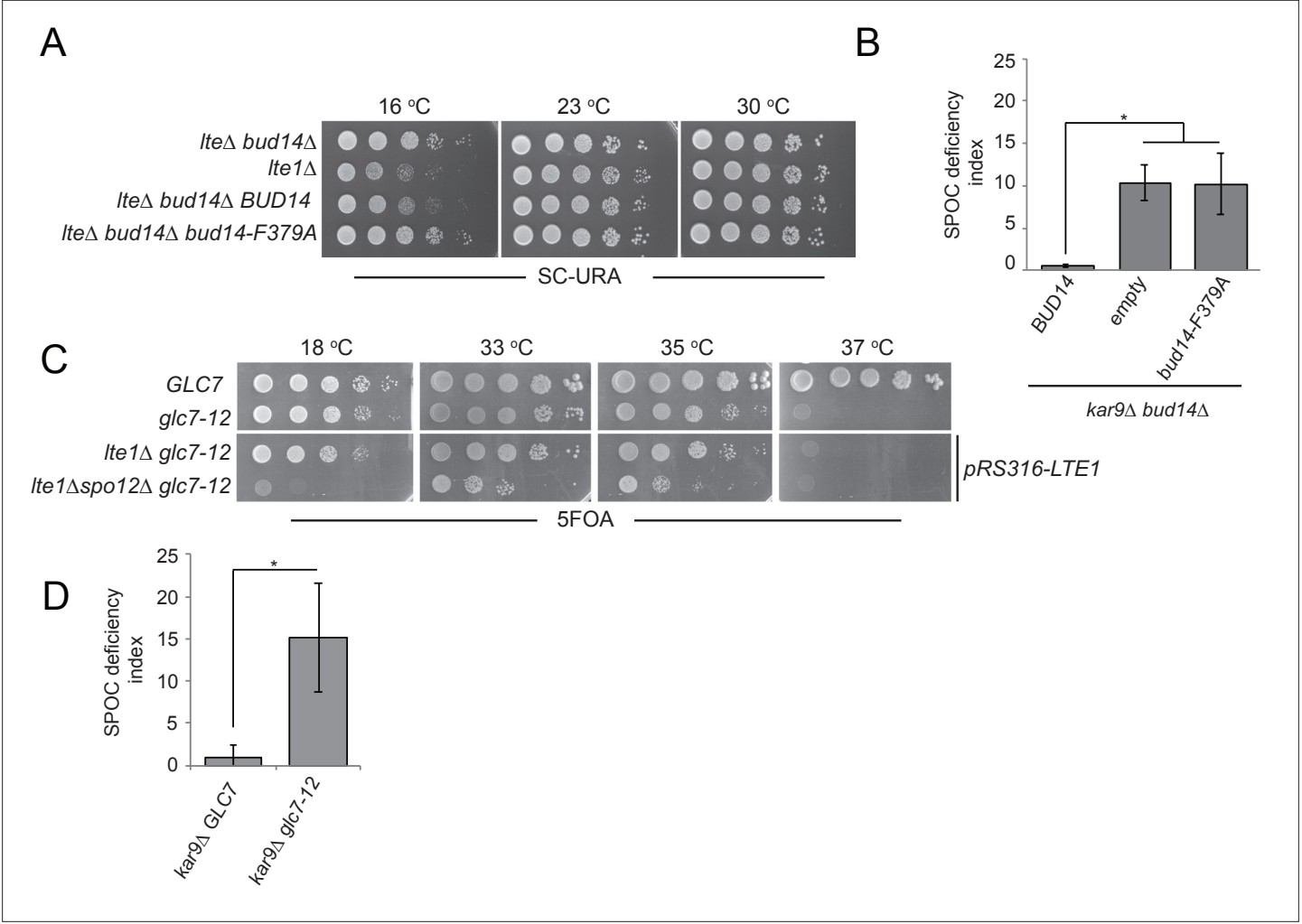

**Figure 4.** Glc7-Bud14 interaction is required for spindle position checkpoint (SPOC) regulatory function of Bud14. (**A**) Serial dilutions of indicated strains bearing *URA3*-based empty plasmid (not indicated on figure) or *BUD14*-containing *URA3*-based plasmids (*BUD14* and *bud14-F379A*) were spotted on SC-URA plate and grown at indicated temperatures. (**B**) SPOC deficiency indexes of indicated strains carrying *URA3*-based empty plasmid (empty) or *BUD14*-containing *URA3*-based plasmids (*BUD14* and *bud14-F379A*). Graphs are average of three independent experiments. A minimum of 100 cells were counted from each strain in each experiment. Error bars show standard deviation. *p<0.05 according to Student's t-test. (**C**) Serial dilutions of indicated strains were spotted on 5-fluoroorotic acid (5-FOA) plate and grown at indicated temperatures. Cells that contain *LTE1* on a *URA3*-based plasmid are indicated with *pRS316-LTE1*. (**D**) SPOC deficiency indexes of indicated strains. *p<0.05 according to Student's t-test.

The online version of this article includes the following figure supplement(s) for figure 4:

**Source data 1.** Numerical data and statistics for *Figure 4B*.

**Source data 2.** Numerical data and statistics for *Figure 4D*.

**Figure supplement 1.** Assessment of *glc7-12* inactivation at different temperatures.

examining other phenotypes attributed to Glc7 deficiency (**Andrews and Stark, 2000**; **Cheng and Chen, 2010**), such as sensitivity to hydroxyurea (**Figure 4—figure supplement 1A**, HU plates) and accumulation of large-budded cells with single nucleus (**Figure 4—figure supplement 1B**). At 18°C, which is a permissive temperature for *glc7-12*, *glc7-12 lte1Δ spo12Δ* cells failed to form colonies (**Figure 4C**), replicating the lethality of *lte1Δ spo12Δ* (**Figure 1A**). However, at the semi-permissive temperatures of *glc7-12* (33–35°C), *glc7-12 lte1Δ spo12Δ* cells were able to form colonies (**Figure 4C**). Thus, similar to the deletion of *BUD14*, partial inactivation of Glc7 rescues lethality of *lte1Δ spo12Δ*. Furthermore, *glc7-12 kar9Δ* cells were SPOC deficient at 33°C (**Figure 4D**). Thus, we conclude that Glc7-Bud14 activity is crucial for SPOC function.

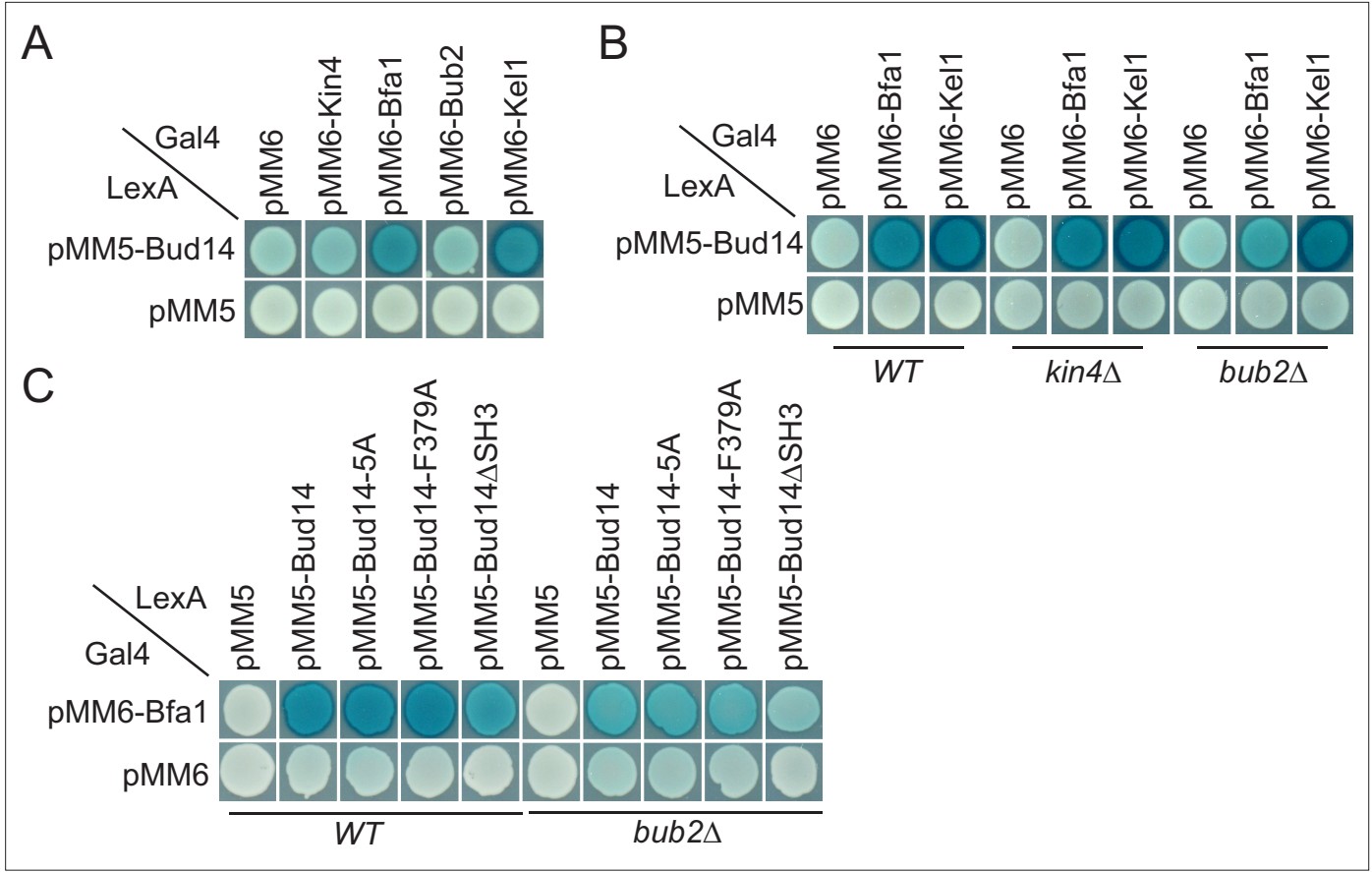

**Figure 5.** Yeast two hybrid analysis of Bud14 and Bud14 mutants with spindle position checkpoint (SPOC) proteins. (**A**) Bud14 interacts with Bfa1 but not with Bub2 or Kin4. (**B**) Bud14-Bfa1 interaction is dependent on Bub2. (**C**) Bfa1-Bud14 interaction is reduced in Bud14ΔSH3 mutant. SGY37 was co-transformed with indicated plasmids. Empty plasmids served as a control for any self-activation. Kel1 served as a positive control. Cells were grown for 2 days on selective agar plates before overlay. Blue color formation was monitored as an indication of protein-protein interaction.

The online version of this article includes the following figure supplement(s) for figure 5:

**Figure supplement 1.** Analysis of Glc7 interaction with spindle position checkpoint (SPOC) components.

**Figure supplement 2.** SH3 domain of Bud14 is important for Bud14 function in spindle position checkpoint (SPOC).

**Figure supplement 2—source data 1.** Numerical data for *Figure 5—figure supplement 2A*.

## Bud14 interacts with Bfa1 in the yeast two hybrid system

We sought to understand the target of Bud14 among SPOC proteins. For this, we performed a yeast two hybrid assay. We used Bfa1, Bub2, and Kin4 as bait proteins and Kel1 as a positive control in this assay. Formation of a blue color indicates activation of the ß-galactosidase gene and hence physical interaction of LexA DNA-binding domain fused with Bud14 (PMM5-Bud14) and Gal4-activation domain fused with the bait protein (pMM6-Bait protein). We observed blue color formation in cells bearing pMM5-Bud14 and pMM6-Kel1 as expected due to their known physical interaction (*Gould et al., 2014*; *Krogan et al., 2006*; *Neller et al., 2015*). We observed a similar interaction between Bud14 and Bfa1 but not Bub2 or Kin4 (*Figure 5A*). We could not, however, detect an interaction between Glc7 and Bfa1, Bub2 or Kin4 in our yeast two hybrid system (*Figure 5—figure supplement 1*). Intriguingly, deletion of *BUB2* but not *KIN4* ceased blue color formation in cells with pMM5-Bud14 and pMM6-Bfa1, indicating that Bud14-Bfa1 interaction is dependent on the presence of Bub2 (*Figure 5B*).

We next analyzed the interaction of Bfa1 with mutant forms of Bud14. Bud14-F379A that cannot bind Glc7 (*Figure 3—figure supplement 1B*; *Knaus et al., 2005*) interacted with Bfa1 similarly to the wild-type Bud14 in our yeast two hybrid setup (*Figure 5C*). Bud14 formin-binding mutant (Bud14-5A) also interacted with Bud14 equally well (*Figure 5C*). Bud14 contains an SH3 domain at its amino

terminal (amino acids 262–318), deletion of which diminishes Bud14 bud cortex localization and also interferes with Bud14-Glc7 interaction (*Knaus et al., 2005*). Notably, deletion of Bud14 SH3 domain also caused a decrease in Bud14-Bfa1 interaction (*Figure 5C*). Consequently, *bud14-SH3Δ* was SPOC deficient (*Figure 5—figure supplement 2A,Figure 3—figure supplement 1*) and rescued *lte1Δ* cold sensitivity (*Figure 5—figure supplement 2B*). Thus, SH3 domain of Bud14 is important not only for Bud14 cortex localization but also for Bud14-Glc7 and Bud14-Bfa1 interaction and hence for its function in SPOC.

## Bud14 limits the amount of Bfa1 at dSPBs but does not affect Bfa1 symmetry upon spindle mispositioning

So far, we showed that Bud14 has a role in SPOC through its interaction with Glc7. We also showed that it interacts with Bfa1. We next asked whether Bud14-Glc7 affects Bfa1 function. SPOC functionality relies on a radical change in SPB localization of Bfa1-Bub2. When the spindle is properly oriented, Bfa1-Bub2 preferentially localizes to the SPB that moves to the daughter cell (dSPB). Upon spindle misalignment, Bfa1-Bub2 is released from dSPB, hence a low level of Bfa1-Bub2 is attained at both SPBs. Thus, we analyzed the effect of Bud14 on SPB localization of Bfa1-Bub2. First, we compared the SPB-bound levels of Bfa1-GFP at SPBs in wild-type and *bud14Δ* cells with normally aligned anaphase spindles (spindle length >3 µm) (*Figure 6A*). To our surprise, more Bfa1-GFP was localized at dSPBs in *bud14Δ* cells compared to wild-type cells. Importantly, increased Bfa1-GFP signal at the dSPB of *bud14Δ* cells returned to wild-type levels when *BUD14* was introduced in these cells on a plasmid (*Figure 6B*). Unlike *BUD14*, introduction of the *bud14-F379A* in *bud14Δ* cells did not cause a significant reduction in the levels of Bfa1 at the dSPB (*Figure 6C*). These data indicate that the presence of Glc7-Bud14 reduces the dSPB-bound Bfa1 levels.

Bub2 and Tem1 localization at SPBs also depend on Bfa1 (*Pereira et al., 2000*). Accordingly, Bub2-GFP and Tem1-GFP localization at dSPB was also remarkably higher in *bud14Δ* cells than wild-type cells (*Figure 6—figure supplement 1A and B*). Such increase in SPB-bound levels of Bfa1-Bub2 and Tem1 was not restricted to anaphase but was also present before anaphase onset (spindle length = 1.5–2 µm; *Figure 6—figure supplement 1C*).

We next asked whether the increased Bfa1-Bub2 and Tem1 recruitment to the dSPBs observed in *bud14Δ* cells may cause premature activation of the MEN in these cells. Mitotic exit is preceded by SPB localization of Cdc15 and Dbf2-Mob1 (*Frenz et al., 2000*; *Luca et al., 2001*; *Pereira et al., 2002*; *Visintin and Amon, 2001*; *Yoshida and Toh-e, 2001*). We assessed MEN activity using the SPB-bound levels of Mob1-GFP owing to its brighter fluorescence signal (*Caydasi et al., 2012*). Mob1-GFP did not prematurely localize to SPBs in *bud14Δ* cells (*Figure 6—figure supplement 2A–C*). The levels of Mob1-GFP at the SPBs of *bud14Δ* cells were not altered in anaphase either (*Figure 6—figure supplement 2D and E*). Thus, we conclude that despite the elevated levels of Bfa1-Bub2 and Tem1 at the dSPB, MEN activity is not enhanced by *BUD14* deletion in cells with normally aligned spindles.

We next analyzed SPB-bound Bfa1 levels in cells with misaligned anaphase spindles. Similarly to the situation in cells with correctly aligned spindles, *bud14Δ kar9Δ* cells with misoriented spindles had more Bfa1-GFP at their dSPBs than *kar9Δ* cells (*Figure 6—figure supplement 1D*). Thus, more Bfa1 localize at the dSPB in the absence of Bud14 regardless of the spindle position and the mitotic stage.

When the anaphase spindle elongates in the wrong direction, phosphorylation of Bfa1 by Kin4 causes dissociation of Bfa1-Bub2 from dSPB. Consequently, upon spindle mispositioning, SPB localization of Bfa1-GFP changes from asymmetric (more at the dSPB than mSPB) to more symmetric (similar at both SPBs). We thus asked whether *bud14Δ* cells were able to change their Bfa1 localization from asymmetric to symmetric in response to spindle misalignment. As a measure of signal asymmetry at SPBs, we calculated Bfa1-GFP asymmetry index as the ratio of Bfa1-GFP intensity at SPBs of *kar9Δ* cells with correctly aligned and misaligned anaphase spindles (*Figure 6D*). In response to spindle mispositioning, Bfa1-GFP asymmetry was drastically reduced in *kar9Δ* cells, whereas it did not significantly change in *kin4Δ kar9Δ* cells (*Figure 6D*). Similarly to *kar9Δ* cells, Bfa1-GFP asymmetry was also greatly decreased in *bud14Δ kar9Δ* cells when the spindle was mispositioned (*Figure 6D*). We also analyzed the behavior of Bfa1-GFP through time-lapse experiments. Consistent with previous reports, SPB-bound levels of Bfa1-GFP in *kar9Δ* cells drastically decreased within a few minutes after elongation of the anaphase spindle in the wrong direction (*Caydasi and Pereira, 2009 Figure 6—figure supplement 3A*). Bfa1-GFP also rapidly decreased at SPBs of *bud14Δ kar9Δ* cells (*Figure 6—figure*

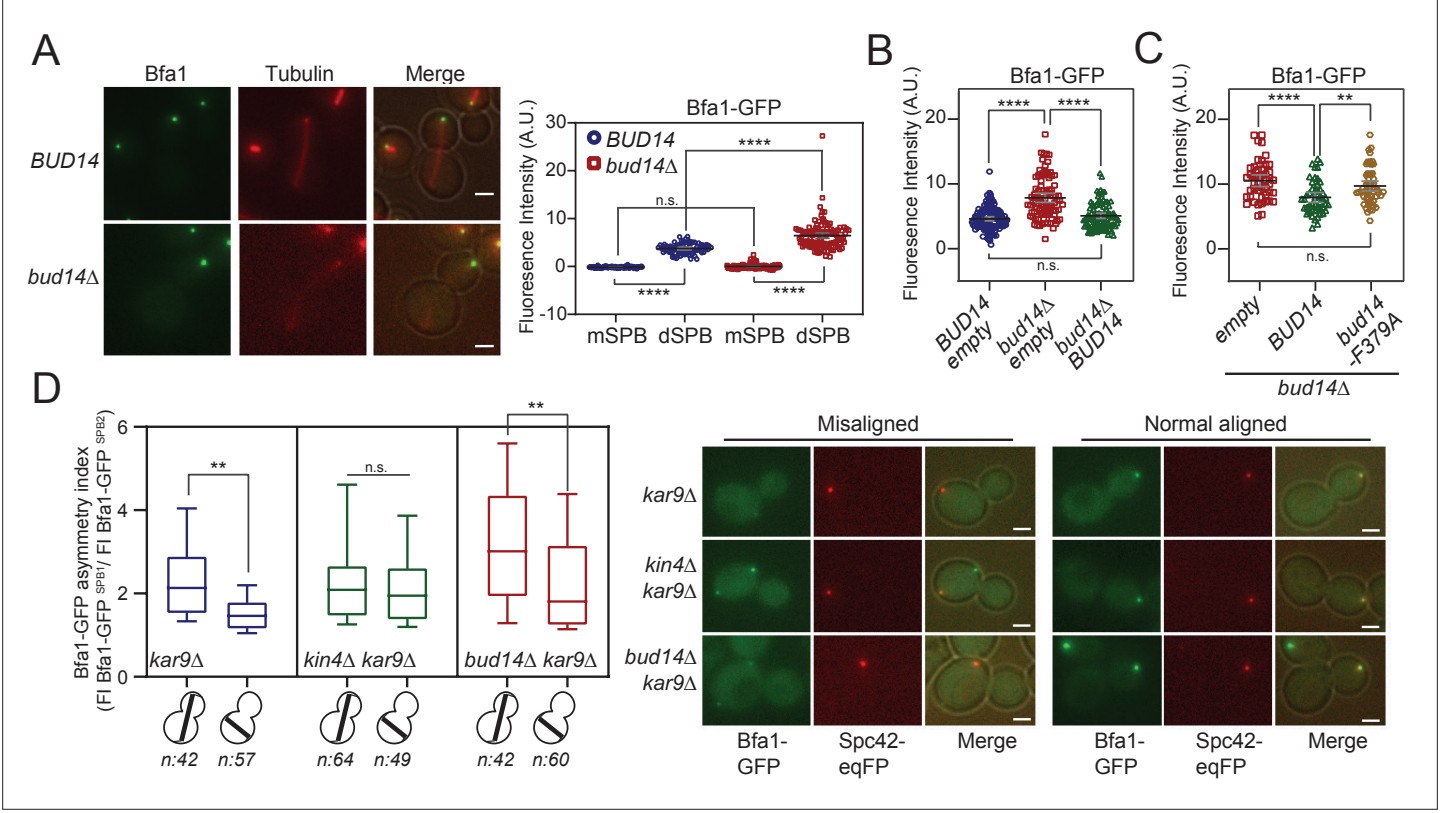

**Figure 6.** More Bfa1 localizes to daughter spindle pole body (dSPB) in the absence of *BUD14*. (**A**) Bfa1-GFP signal intensity quantifications at the SPBs of *BFA1-GFP mCherry-TUB1 or BFA1-GFP mCherry-TUB1 bud14Δ* cells with normal aligned anaphase spindles were plotted on the right. Black and gray lines in the dotplots are the mean value and standard deviation, respectively. Representative still images of cells are shown on the left. (**B**) Bfa1-GFP signal intensities at the dSPB were plotted in *BUD14* and *bud14Δ* cells bearing empty plasmid (blue and red), as well as *bud14Δ* cells bearing a ADH-*BUD14*-containing plasmid (green). (**C**) Bfa1-GFP signal intensities at the dSPB were plotted in *bud14Δ cells* bearing empty plasmid (red), ADH1-*BUD14*-containing plasmid (green) and ADH1-*bud14-F379A*-containing plasmid (yellow) (**D**) Analysis of Bfa1-GFP asymmetry at the SPBs of *kar9Δ*, *kar9Δ kin4Δ*, and *kar9Δ bud14Δ* cells with correctly aligned and misaligned anaphase spindles. Box and Whisker plot shows the ratio of Bfa1-GFP signal intensities at the SPB1 and SPB2, where SPB1 always corresponds to SPB with the greater Bfa1-GFP signal. The box represents first and third quartile while the whiskers show 10–90 percentile of the data. The horizontal line in the box indicates the median of the data. Only comparisons of normal and misaligned spindles are shown in the figure. Representative still images of cells are shown on the right. n: sample size. Scale bar: 2 µm. One-way ANOVA with uncorrected Fisher's LSD was applied for all statistical analyses. **p<0.01, ***p<0.001, ****p<0.0001. All pairwise comparisons are shown in the corresponding source data files.

The online version of this article includes the following figure supplement(s) for figure 6:

**Source data 1.** Numerical data and statistics for *Figure 6A*.

**Source data 2.** Numerical data and statistics for *Figure 6B*.

**Source data 3.** Numerical data and statistics for *Figure 6C*.

**Source data 4.** Numerical data and statistics for *Figure 6D*.

**Figure supplement 1.** Bfa1, Bub2, and Tem1 localization in bud14Δ cells during anaphase and metaphase.

**Figure supplement 1—source data 1.** Numerical data and statistics for *Figure 6—figure supplement 1A*.

**Figure supplement 1—source data 2.** Numerical data and statistics for *Figure 6—figure supplement 1B*.

**Figure supplement 1—source data 3.** Numerical data and statistics for *Figure 6—figure supplement 1C*.

**Figure supplement 1—source data 4.** Numerical data and statistics for *Figure 6—figure supplement 1D*.

**Figure supplement 2.** Analysis of Mob1 spindle pole body (SPB) localization.

**Figure supplement 2—source data 1.** Numerical data and statistics for *Figure 6—figure supplement 2C*.

**Figure supplement 2—source data 2.** Numerical data and statistics for *Figure 6—figure supplement 2D*.

**Figure supplement 3.** Bfa1-GFP localization during spindle misalignment.

*supplement 3B*), which was not the case for *kar9Δ kin4Δ* cells (*Figure 6—figure supplement 3C*).

Thus, unlike Kin4, Bud14 does not promote a change in SPB localization of Bfa1 in response to spindle mispositioning, but rather limits the levels of SPB-bound Bfa1 regardless of the spindle position. These observations altogether suggest a role for Bud14-Glc7 in controlling Bfa1 sequestration at the SPB in a way differently than Kin4 and further support the conclusion that Kin4 and Bud14-Glc7 work in different branches of SPOC activation.

## Glc7-Bud14 promotes Bfa1 dephosphorylation

Bfa1 phosphorylation is a key regulator of mitotic exit. Cdc5 phosphorylation of Bfa1 inhibits Bfa1-Bub2 GAP activity towards Tem1 to promote mitotic exit, whereas Kin4 phosphorylation of Bfa1 prevents Cdc5 phosphorylation of Bfa1 to block mitotic exit (*Geymonat et al., 2002*; *Hu et al., 2001*; *Maekawa et al., 2007*). We asked whether Bud14 impinges on Bfa1 phosphorylation. To investigate Bfa1 phosphorylation, we made use of *lte1Δ* Gal1-UPL-*TEM1* cells that arrest in late anaphase upon depletion of Tem1 with Kin4 localized at both SPBs (*Bertazzi et al., 2011*; *Hu et al., 2001*). This arrest mimics a SPOC-activated state where Cdc5 phosphorylation of Bfa1 is prevented by Kin4 phosphorylation of Bfa1 and consequently Bfa1 hyperphosphorylation is restored upon deletion of *KIN4* in these cells (*Bertazzi et al., 2011*; *Figure 7—figure supplement 1*). Markedly, deletion of *BUD14* also resulted in appearance of slow-migrating Bfa1 forms in *lte1Δ* Gal1-UPL-*TEM1* cells (*Figure 7A and B*, *Figure 7—figure supplement 1B*), although Kin4 was localized at both SPBs (*Figure 2—figure supplement 1D*). Such slow-migrating forms of Bfa1 in *lte1Δ bud14Δ* cells were dependent on Cdc5 as they disappeared in *cdc5-10 lte1Δ bud14Δ* cells arrested in anaphase (*Figure 7C*). These data indicate that Bud14 prevents Cdc5-dependent hyperphosphorylation of Bfa1 in *lte1Δ* cells.

We next analyzed the effect of *BUD14* overexpression on Bfa1 phosphorylation. Bud14 overexpression causes cells to arrest in a pre-anaphase state, and, hence, it is lethal (*Chesarone et al., 2009*; *Cullen and Sprague, 2002*; *Knaus et al., 2005*; *Pinsky et al., 2006*). Of importance, this lethality is not dependent on Bfa1 or Kin4 (*Figure 7—figure supplement 2A*). To obtain cells arrested in anaphase in the presence of high levels of Bud14, we first arrested *cdc15-as* (L99G) (*Bishop et al., 2000*; *D'Aquino et al., 2005*) bearing cells in anaphase through 1NM-PP1 treatment and then induced Gal1-*BUD14* overexpression by addition of galactose into the growth medium. Overexpression of *BUD14* (*Figure 7D*) but not *bud14-F379A* caused a reduction in hyperphosphorylated forms of Bfa1 (*Figure 7E and F*). This data suggests that Glc7-Bud14 causes dephosphorylation of hyperphosphorylated Bfa1 in anaphase-arrested cells.

To further understand whether Glc7-Bud14 dephosphorylates Bfa1, we performed an in vitro dephosphorylation assay. For this, we enriched Bfa1-3HA from Gal1-UPL-*TEM1 kin4Δ* cells arrested in telophase through depletion of Tem1 and incubated it with or without immunoprecipitated Glc7-TAP in the presence or absence of 1.5 μM okadaic acid to inhibit Glc7 (*Bialojan and Takai, 1988*; *Cohen et al., 1989a*; *Cohen et al., 1989b*; *Figure 7G*). Addition of Glc7-TAP in the phosphatase reaction caused a downshift of Bfa1-3HA, but not when okadaic acid was added (*Figure 7G*). Same behavior was observed when Bud14-TAP was used in the phosphatase reaction instead of Glc7-TAP (*Figure 7—figure supplement 2B*). These data altogether show that Glc7-Bud14 promotes dephosphorylation of hyperphosphorylated Bfa1 in anaphase.

## Discussion

SPOC is a surveillance mechanism that inhibits mitotic exit in response to spindle mispositioning. Protein phosphorylation and localization play a key role in SPOC. Especially the activity of Bfa1, the most downstream effector of the SPOC, is tightly controlled with respect to the spindle orientation through phosphorylation by the polo like kinase Cdc5 and the mother-specific kinase Kin4 in an opposing manner. Phosphorylation of Bfa1 in turn affects activity and localization of Bfa1-Bub2 and hence the activity and the localization of the GTPase Tem1 at SPBs. Despite the knowledge on Bfa1 phosphorylating kinases, knowledge on phosphatases counteracting these kinases has been limited. In this study, we established Glc7-Bud14 as a novel SPOC component that promotes dephosphorylation of Bfa1 when Bfa1 is hyperphosphorylated in anaphase in a Cdc5-dependent manner. Our work supports a model where Bfa1-Bub2 is activated by Glc7-Bud14 in parallel to Kin4. Upon spindle misalignment in this model, the SPOC kinase Kin4 phosphorylates Bfa1 to remove Bfa1 from the SPBs,

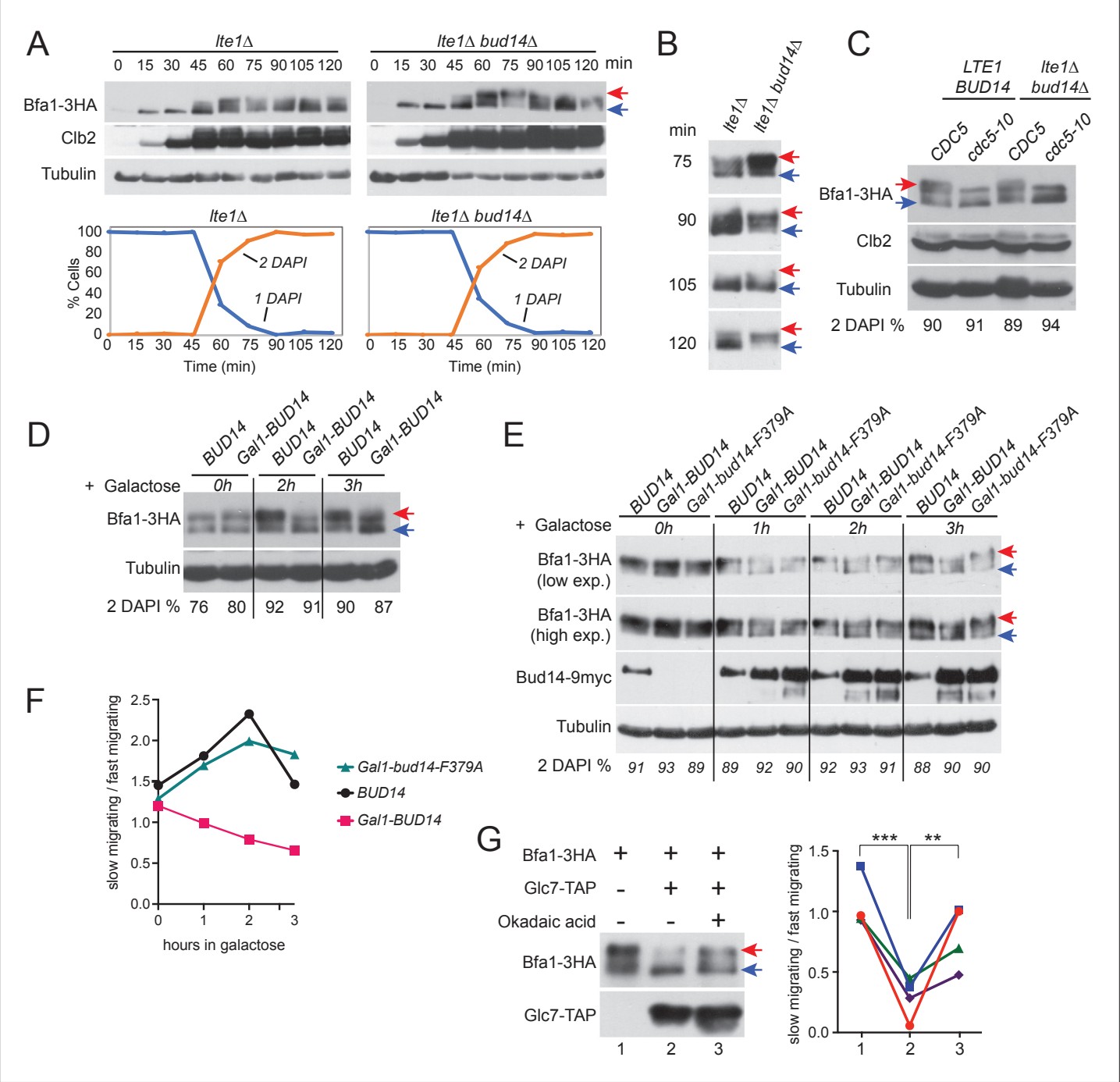

**Figure 7.** Glc7-Bud14 promotes Bfa1 dephosphorylation in anaphase. (**A**) Bfa1-3HA Gal1-UPL-Tem1-containing *lte1Δ* and *lte1Δ bud14Δ* cells grown in galactose-containing medium were released from alpha factor-imposed G1 arrest (t = 0) into an alpha factor-free medium supplemented with glucose to achieve Tem1 depletion, and samples were collected at indicated time points. Bfa1-3HA mobility shift was analyzed via western blotting using anti-HA antibodies. Red arrow indicates hyperphosphorylated forms of Bfa1-3HA, whereas blue arrow indicates hypophosphorylated forms of Bfa1-3HA. Percentage of cells with single nucleus (1 DAPI) and two separate nuclei (2 DAPI) were plotted as a marker for cell cycle progression. (**B**) Indicated time points of each cell type from the experiment shown in (**A**) were loaded side-by-side for better comparison of Bfa1-3HA mobility. (**C**) Bfa1-3HA mobility in Gal1-UPL-*TEM1* or *cdc5-10*-bearing *LTE1 BUD14* or *lte1Δbud14Δ* cells. Percentage of cells with two separate nuclei (% 2 DAPI) are indicated as a measure of cells in anaphase. Cells were first arrested in G1, then released from the G1 arrest and cultured for 90 min before sample collection. Gal1-UPL-Tem1 cells were treated as in (**A**) to achieve the anaphase arrest, whereas anaphase arrest of cdc5-10 cells was achieved through growth at 37°C. (**D**) *cdc15-as*-bearing cells bearing *BUD14* or Gal1-*BUD14* at the Bud14 endogenous locus were grown to log-phase in raffinose-containing medium, treated with 1NM-PP1 for 2,5 hr followed by galactose addition (t0) to the medium. Samples were collected at 0 hr, 2 hr, and 3 hr after galactose addition

*Figure 7 continued on next page*

*Figure 7 continued*

and Bfa1 mobility was analyzed. Percentage of cells with two separate nuclei (% 2 DAPI) is indicated as a measure of cells in anaphase. (**E**) *cdc15-as bud14Δ* cells with *BUD14*-9myc, Gal1-*BUD14*-9myc, or Gal1-*bud14-F379A*-9myc integrated at the chromosomal leu2 locus were grown to log-phase in raffinose-containing medium, treated with 1NM-PP1 for 3 hr, followed by galactose addition (t0) to the medium. Samples were collected at 0 hr, 1 hr, 2 hr, and 3 hr after galactose addition and Bfa1 mobility was analyzed. Percentage of cells with two separate nuclei (% 2 DAPI) is indicated as a measure of cells in anaphase. (**F**) Quantification of relative levels of hypersphosphorylated Bfa1 from the experiment shown in (**E**). Band intensity ratio of slow-migrating forms to fast-migrating forms of Bfa1 is plotted. (**G**) In vitro phosphatase assay of immunoprecipitated Glc7-TAP on IgG beads is incubated with Bfa1-3HA purified from *BFA1-3HA* Gal1-*UPL-TEM1 kin4Δ* cells in the presence or absence of 1.5 μM okadaic acid. As a no Glc7-TAP control, IgG beads incubated with cell lysates of ESM356-1 were used. Glc7-TAP levels were detected using anti-TAP antibodies. Bfa1-3HA was detected using anti-HA antibodies. Red and blue arrows indicate slow-migrating and fast-migrating forms of Bfa1-3HA, respectively. Quantification of relative levels of hypersphosphorylated Bfa1 is shown on the right. Each color represents a different independent experiment. One-way ANOVA with uncorrected Fisher's LSD was applied for statistical analysis. **$p < 0.01$, ****$p < 0.0001$.

The online version of this article includes the following source data and figure supplement(s) for figure 7:

**Source data 1.** Labeled uncropped blot images for *Figure 7A and B*.

**Source data 2.** Raw scans of the x-ray films for *Figure 7—source data 1b*, (**A**) anti-HA blot, (**B**) anti-Clb2 blot, (**C**) anti-tubulin blot, and (**D**) anti-HA blot.

**Figure supplement 1.** Analysis of Bfa1 mobility in lte1Δ and lte1Δ kin4Δ cells.

**Figure supplement 1—source data 1.** Labeled uncropped blot images for *Figure 7—figure supplement 1*.

**Figure supplement 1—source data 2.** Raw scans of the blot images for (**A**) anti-HA blot.

**Figure supplement 2.** Kin4 overexpression toxicity in *bud14Δ* cells and phosphatase assay using Bud14-TAP.

**Figure supplement 2—source data 1.** Labeled uncropped blot images for *Figure 7—figure supplement 2B*.

**Figure supplement 2—source data 2.** Raw scans of the blot images for (**A**) anti-HA blot and (**B**) anti-TAP and anti-Myc blots.

**Source data 3.** Labeled uncropped blot images for (**A**) *Figure 7E and G*.

**Source data 4.** Raw scans of the x-ray films for (**A**) anti-HA blot, (**B**) anti-Myc blot, upper panel of the scan, (**C**) anti-tubulin blot, mid-panel of the scan, (**D**) anti-HA blot, left panel of the scan, and (**E**) anti-TAP blot.

---

away from the inhibitory kinase Cdc5, and Glc7-Bud14 dephosphorylates Bfa1 to remove its inhibitory phosphorylation.

## Glc7-Bud14 is a SPOC component

Our study identified Glc7, the budding yeast protein phosphatase 1, in association with its regulatory subunit Bud14 as a novel SPOC component that inhibits mitotic exit upon spindle mispositioning. Several lines of evidence led us to this conclusion. First, similar to other SPOC proteins, *BUD14* deletion rescued growth of several mutants impaired in mitotic exit, including the cold-sensitive *lte1Δ* cells, the temperature-sensitive MEN mutants (*cdc15-1*, *dbf2-2*, and *mob1-67*), and the synthetic lethal *lte1Δ spo12Δ* cells. Partial impairment of Glc7 function via the *glc7-12* mutant also rescued growth lethality of *lte1Δ spo12Δ* cells. In addition, a Bud14 mutant that cannot bind Glc7 (*bud14-F379A*) (*Knaus et al., 2005*) also rescued cold sensitivity of *lte1Δ* cells. Second, similar to other SPOC proteins, *BUD14* deletion caused accumulation of multinucleated cells when spindle misalignment was induced in cells by deletion of *KAR9* or *DYN1*, implying that *bud14Δ* cells are deficient in arresting upon mispositioning of the anaphase spindle. Of importance, similar to *bud14Δ*, *glc7-12* and the *bud14-F379A* mutants were also SPOC deficient. Bud14 was implicated in the Dynein pathway of spindle positioning (*Knaus et al., 2005*). However, deletion of *BUD14* caused accumulation of multinucleated phenotypes not only in *kar9Δ* but also in *dyn1Δ* backgrounds, indicating that the function of Bud14 in Dynein activation is not contributing to the observed multinucleation phenotype. In line with this notion, time-lapse analysis revealed that *bud14Δ kar9Δ* cells exited mitosis regardless of the spindle position. Furthermore, similar to other SPOC components, absence of *BUD14* did not affect the timing of mitotic exit when the spindle was correctly positioned but was crucial for delaying exit from mitosis when the spindle is misoriented. Data supporting this included the anaphase duration of *bud14Δ kar9Δ* cells with normally positioned spindles, which was not different than that of *kar9Δ* cells and also Mob1-GFP localization in *bud14Δ* cells, which was not different than that in wild-type cells.

## Glc7-Bud14 works in a pathway parallel to Kin4

Our data showed that Bud14 does not work in the Kin4 branch of SPOC. First, Kin4 localization and phosphorylation did not depend on Bud14. Second, Kin4 overexpression still caused toxicity in

*BUD14*-deleted cells, indicating that Kin4, when produced at high levels, is able to keep Bfa1 active. Furthermore, time-lapse analysis of Bfa1 localization during spindle misalignment showed that Bfa1 is being removed from SPBs of *bud14Δ* cells with misaligned spindles, while removal of Bfa1 from SPBs is largely a Kin4-dependent process (*Caydasi and Pereira, 2009*; *Monje-Casas and Amon, 2009*). Thus, Kin4 is able to exert its effect on Bfa1 in the absence of *BUD14*. These data altogether exclude the possibility that Bud14 works through Kin4 branch of the SPOC and suggests a model in which Glc7-Bud14 prevents mitotic exit of cells with misaligned spindles in a way parallel to Kin4. In support of this model, we observed an additive decrease in anaphase duration of *kin4Δ bud14Δ* cells with misaligned spindles compared to the cells bearing single-gene deletions of *KIN4* or *BUD14*. Importantly, *kin4Δ bud14Δ* cells completely failed to delay mitotic exit while *kin4Δ* or *bud14Δ* cells were able to slightly delay mitotic exit in response to spindle misorientation (*Figure 2C*). Accordingly, SPOC activity in *kin4Δ bud14Δ* cells was as disrupted as in *bfa1Δ* cells (*Figure 2A*). Thus, we propose that Kin4 and Bud14 work in parallel and presence of both is critical for a robust SPOC arrest.

Interestingly, neither *KIN4* nor *BUD14* is required for spindle assembly checkpoint-dependent metaphase arrest of cells, whereas Bfa1-Bub2 is (data not shown; *Pereira and Schiebel, 2005*). This suggests that Kin4 and Bud14 become important to halt mitotic exit after APC/Cdc20 activation at the metaphase-anaphase transition. Anaphase onset also brings about the partial activation of Cdc14 by the FEAR pathway, which primes MEN activation in several ways, and Kin4 was shown to be dispensable for SPOC in cells lacking FEAR activity (*Caydasi et al., 2017*; *Falk et al., 2016a*; *Scarfone et al., 2015*). Consistently, Bud14 was also dispensable for SPOC in *spo12Δ* cells that lack FEAR (data not shown). These data support that Kin4 and Bud14 work in parallel to keep Bfa1-Bub2 active in anaphase, to provide an arrest of anaphase cells prior to mitotic exit.

## Glc7-Bud14 promotes dephosphorylation of Bfa1

The SPOC component Bfa1 is a highly phosphorylated protein in a cell cycle-dependent manner. So far, Bfa1 has been shown to be the target of kinases Cdk1, Cdc5, and Kin4 (*Caydasi et al., 2017*; *Geymonat et al., 2003*; *Hu et al., 2001*; *Pereira and Schiebel, 2005*). Among these kinases, Cdc5 plays an inhibitory role, whereas Cdk1 and Kin4 play an activating role. It is possible that other kinases may also phosphorylate Bfa1. There has been an ongoing curiosity for phosphatases that act on Bfa1. So far, Cdc14 was shown to counteract Cdk1 phosphorylation of Bfa1 to partially inhibit Bfa1-Bub2 activity after anaphase onset (*Caydasi et al., 2017*) and PP2A-Cdc55 was shown to counteract Cdc5-dependent phosphorylation of Bfa1 before anaphase onset, and hence downregulation of PP2A-Cdc55 activity after anaphase onset facilitates Bfa1 inhibition (*Baro et al., 2013*). Our data that Bfa1 appears hyperphosphorylated in *lte1Δ bud14Δ* cells, in a Cdc5-dependent manner, during anaphase indicate that Glc7-Bud14 promotes dephosphorylation of hyperphosphorylated Bfa1 after anaphase onset. Consistently, overexpression of *BUD14* but not the *bud14-F379A*, which cannot bind Glc7, promoted dephosphorylation of Bfa1 in anaphase-arrested cells. Furthermore, through an in vitro dephosphorylation assay we showed that Glc7-Bud14 is able to dephosphorylate hyperphosphorylated Bfa1 purified from anaphase-arrested cells.

We further showed that Bfa1 and Bud14 interacted in yeast two hybrid assays. However, it is important to note that despite the observed yeast two hybrid-based interaction between Bfa1 and Bud14, and observed in vitro enzymatic activity of Glc7-Bud14 towards hyperphosphorylated Bfa1, we failed to detect Bfa1-Bud14 physical interaction via co-immunoprecipitation-based methods. Although this may indicate that only a minor pool of Bfa1 and Bud14 may be interacting in a very transient manner, further studies will be necessary to show the nature of this interaction. Of note, Bfa1-Bud14 interaction observed in the yeast two hybrid assays required presence of Bub2. This data not only rules out the possibility of a false-positive Bfa1-Bud14 yeast two hybrid interaction in wild-type cells, but also suggest that Bud14 may recognize Bfa1-Bub2 as a complex, or the interaction between Bud14 and Bfa1 may require in vivo modifications of Bfa1 that occur only when bound to Bub2. It has been known that hyperphosphorylated forms of Bfa1 abolish in the absence of Bub2 (*Lee et al., 2001b*). In addition, Bfa1 fails to localize to SPBs in the absence of Bub2 (*Pereira et al., 2000*). Furthermore, Bfa1 phosphorylation by Cdc5 takes place at SPBs (*Maekawa et al., 2007*; *Pereira et al., 2000*).

These findings, altogether, are suggestive of a model in which Glc7-Bud14 may recognize specifically a phosphorylated form of Bfa1. We favor a speculative model where Glc7-Bud14 dephosphorylates

Cdc5-phosphorylated Bfa1 specifically upon spindle misalignment. This model is also in concordance with our previous conclusion that Glc7-Bud14 works in a parallel pathway to Kin4. Removal of Cdc5-dependent phosphorylation of Bfa1 by Glc7-Bud14 and isolation of Bfa1 away from the inhibitory Cdc5 kinase activity by Kin4 would concertedly reduce the levels of Cdc5-phosphorylated Bfa1 and thus work together to rapidly increase the levels of active Bfa1-Bub2.

Kin4 phosphorylation of Bfa1 prevents Cdc5 phosphorylation of Bfa1 both during SPOC activation (i.e., anaphase arrest of *lte1Δ* cells that mimics SPOC arrest) and during the metaphase arrest upon nocodazole treatment (*Bertazzi et al., 2011*; *D'Aquino et al., 2005*; *Pereira and Schiebel, 2005*). However, unlike *kin4Δ* cells, *bud14Δ* cells arrested in metaphase through nocodazole treatment did not have hyperphosphorylated Bfa1 (data not shown). This discrepancy between Kin4 and Glc7-Bud14 may indicate alternative scenarios: first, Glc7-Bud14 may dephosphorylate sites other than those phosphorylated by Cdc5. Second, there may be redundant mechanisms of Bfa1 dephosphorylation at Cdc5 phosphorylation sites that especially dominate in metaphase. PP2A-Cdc55 was shown to counteract Bfa1 phosphorylation before anaphase onset (*Baro et al., 2013*). Therefore, one possibility is that PP2A-Cdc55 and Glc7-Bud14 may work redundantly to dephosphorylate Cdc5-phosphorylated Bfa1 at different stages of the cell cycle, metaphase, and anaphase, respectively.

In addition, a truncated version of Bud14 that lacks the SH3 domain was also unable to inhibit mitotic exit. This truncated Bud14 is unable to localize at the bud cortex and is also unable to bind Glc7 (*Knaus et al., 2005*; *Figure 3—figure supplement 1B*). Our yeast two hybrid analysis also revealed that Bud14-SH3Δ also have a reduced binding to Bfa1 (*Figure 5C*). Thus, the loss in SPOC activity of the Bud14-SH3Δ can be attributed to its reduced association with Glc7 and Bfa1.

Intriguingly, another version of Bud14, which lacks the formin-regulatory motif (Bud14-5A), was able to yield a stronger interaction with Glc7 in our yeast two hybrid assays (*Figure 3—figure supplement 1B*). This increased association may explain why Bud14-5A provided a stronger SPOC activity than the Bud14 wild type based on SPOC functionality measurements (*Figure 3A*).

## Bud14 deletion causes an increase in dSPB-bound levels of Bfa1-Bub2 and Tem1

We observed that cells lacking *BUD14* had more Bfa1, Bub2, and Tem1 localized at dSPB, making their localization more asymmetric. The asymmetry of Bfa1 at SPBs was shown to be dependent on Cdc5, as in the absence of Cdc5 activity Bfa1 localized more symmetrically to SPBs (*Kim et al., 2012*). Thus, presence of more Cdc5-phosphorylated Bfa1 in *bud14Δ* cells may account for increased levels of Bfa1 at the dSPB, and also Bub2 and Tem1, which bind SPBs in association with Bfa1. Alternatively, Glc7-Bud14 may have other targets at the SPB that might normally decrease Bfa1-Bub2 docking sites at the SPBs.

Studies that used versions of Bfa1 forced to bind SPBs stably (*Scarfone et al., 2015*; *Valerio-Santiago and Monje-Casas, 2011*), suggested that increased Bfa1-Bub2 localization at SPBs can lead to increased MEN activity, as it also causes increased Tem1 association with SPBs. Does altered localization of Bfa1-Bub2 and Tem1 observed in *bud14Δ* contribute to mitotic exit? Analysis of Mob1 localization as a marker for MEN activation suggests that *bud14Δ* cells do not activate MEN prematurely. This suggests that Bud14-Glc7 may be inhibiting mitotic exit by activation of Bfa1-Bub2 GAP complex through counteracting an inactivating phosphorylation of Bfa1 rather than through modulating SPB-bound levels of Bfa1-Bub2. We favor that the change in SPB localization is a consequence of the change in the phosphorylation status of Bfa1 or another target at the SPB.

## Role of PP1 through the cell cycle

PP1 has numerous functions in the control of the cell cycle (*Holder et al., 2019*; *Kim et al., 2016*; *Moura and Conde, 2019*; *Nasa and Kettenbach, 2018*). PP1, together with PP2A, is required for mitotic exit in animal cells by counteracting CDK activity (*Wu et al., 2009*). In human cells, PP1 also regulates centrosome splitting by antagonizing the Nek2 kinase (*Meraldi and Nigg, 2001*). In addition, PP1 facilitates abscission by counteracting Plk1-mediated phosphorylation of the centrosomal protein CEP55 during late mitosis and thus promoting its midbody localization (*Gao et al., 2018*). In meiosis of *Xenopus* oocytes, PP1 regulates G2/M transition by partially activating Cdc25 (*Perdiguero and Nebreda, 2004*). From yeast to mammals, PP1 is essential for silencing of the spindle assembly checkpoint through stabilization of kinetochore microtubule interactions by counteracting Aurora B

and potentially regulating recruitment of MCC to kinetochores (*Kelly and Funabiki, 2009*; *Kotwaliwale and Biggins, 2006*; *Musacchio and Salmon, 2007*; *Ruchaud et al., 2007*). In this study, we show that the budding yeast PP1, Glc7, in association with its regulatory subunit, Bud14, inhibits mitotic exit upon anaphase spindle mispositioning. This function of PP1 seems to be opposite to the mitotic exit, promoting the role of PP1 reported in animal cells. The fact that mitotic exit is achieved by the Cdc14 phosphatase in budding yeast, instead of the PP1 and PP2 phosphatases (*Manzano-López and Monje-Casas, 2020*), is likely the reason for Glc7 gaining a different function after the anaphase onset during the narrow time window until exit from mitosis.

Glc7-Bud14 was implicated in microtubule cortex interactions. Data based on Kin4 activity on Bfa1 indicates that SPOC mechanism may sense microtubule-cortex interactions (*Caydasi and Pereira, 2009*; *Maekawa et al., 2007*; *Monje-Casas and Amon, 2009*; *Pereira and Schiebel, 2005*). How spindle alignment is sensed and this information activates SPOC still remain unknown. Due to their known function in regulating microtubule-cortex interactions (*Knaus et al., 2005*), Glc7-Bud14 constitutes a great candidate for sensing of such interactions. Considering that Glc7 (PP1) is a conserved phosphatase, similar mechanisms may be employed by higher eukaryotes. Future work will show whether Glc7-Bud14 may act as a sensor that translates the unattached microtubule-cortex interactions to the downstream SPOC machinery.

## Materials and methods
### Yeast methods, strains, and growth
All yeast strains used are isogenic with S288C and are listed in *Table 1*. Basic yeast methods and growth media were as described (*Sherman, 1991*). Carbon source is glucose unless otherwise stated. When necessary, 2% D-raffinose, 3% D(+)-galactose was used instead of glucose. Yeast strains were grown at 30°C and in rich medium unless otherwise stated. Plasmid-containing strains were grown in synthetic complete (SC) media lacking the appropriate auxotrophic nutrients. The temperature-sensitive mutants and *kar9Δ* cells were maintained at 23°C. Cassette PCR-based gene editing method was used for chromosomal gene deletion and C-terminal or N-terminal tagging (*Janke et al., 2004*; *Knop et al., 1999*). To determine genetic interactions based on growth, mutant yeast strains bearing the wild-type gene on a *URA3*-based plasmid (pRS316) were spotted on 5-fluoroorotic acid (5-FOA) plates. 5-FOA plates negatively select for the *URA3*-based plasmid and allow for observation of the growth phenotype in the absence of the corresponding plasmid. Most *kar9Δ* cells were maintained complemented with *KAR9* on *URA3-based* centromeric plasmid (pGW399), and stroke on 5-FOA plates to induce plasmid lost shortly before experiments on SPOC analysis. Similarly, MEN-ts mutants were maintained complemented with plasmids carrying the wild-type version of the corresponding MEN gene. The plasmids used in this study are listed in *Table 2*.

### Cell growth comparison on agar plates
Yeast cultures were grown in appropriate media and growth conditions for 24 hr. The $OD_{600}$ of the cultures were adjusted to 1, and 10-fold serial dilutions were made using sterile PBS. 10 µl of serial dilutions were spotted on appropriate agar plates and grown at appropriate temperatures for 1–3 days.

### Cell cycle synchronizations
Cells were synchronized in G1 phase by treating log-phase cultures with 10 µg/ml α-factor (Sigma #T6901) dissolved in DMSO and incubating for one and a half doubling time. For synchronization in metaphase without spindles, 15 µg/ml nocodazole (Sigma #M1404) dissolved in DMSO were added to log-phase cultures and incubated for ~2 hr. The arrest was confirmed by microscopy after fixing the cells with 70% ethanol and resuspending in PBS containing 1 µg/ml 4',6-diamino-2-phenylindole (DAPI, Sigma). For nocodazole arrest from G1 synchronized yeast cultures, cells were first treated with α-factor and released in nocodazole-containing YPAD. For telophase arrest of Gal1-UPL-Tem1-bearing cells, after cultures were grown to log-phase in raffinose-galactose-containing medium, cells were transferred into YPAD medium and grown for ~3 hr until cells (>%90) were arrested with large buds and separated nuclei. For telophase arrest from G1-arrested cells: log-phase cultures of Gal1-UPL-Tem1-bearing cells grown in raffinose/galactose were treated with α-factor. 30 min before G1 synchronization was completed, glucose was added to the culture to give 2% final concentration.

**Table 1.** Yeast strains used in this study.

| Strain name | Description | Reference |
|---|---|---|
| AKY038 | ESM356 Bfa1-GFP-kanMX6 Spc42-eqFP- hphNT1 mCherry-Tub1-URA3 | This study |
| AKY043 | ESM356 BFA1-GFP-kanMX6 Spc42-eqFP-hphNT1 kar9Δ::klTRP1 pRS316-KAR9 | *Caydasi and Pereira, 2009* |
| AKY1533 | ESM356 leu2::LEU2-pGal1-KIN4 | This study |
| AKY1574 | ESM356 leu2::LEU2-pGal1-KIN4 bub2Δ::hphNT1 | This study |
| AKY2526 | FAY145 lte1Δ::kanMX6 pRS316-LTE1 spo12Δ::natNT2 | *Caydasi et al., 2017* |
| AKY260 | ESM356 kar9Δ::HIS3MX6 pRS316-KAR9 | *Caydasi et al., 2017* |
| AKY2916 | FAY145 lte1Δ::kanMX6 pRS316-LTE1 spo12Δ::natNT2 bud14Δ::HIS3MX6 | This study |
| AKY2917 | YPH499 GFP-Tub1-URA3 dyn1Δ::klTRP1 bud14Δ::HIS3MX6 | This study |
| AKY2918 | YPH499 GFP-Tub1-URA3 dyn1Δ::klTRP1 kin4Δ::hphNT1 bud14Δ::HIS3MX6 | This study |
| AKY315 | ESM356 kar9Δ::HIS3MX6 pRS316-KAR9 bfa1Δ::klTRP1 | *Caydasi et al., 2017* |
| AKY321 | ESM356 kar9Δ::HIS3MX6 pRS316-KAR9 kin4Δ::klTRP1 | *Caydasi et al., 2017* |
| AKY346 | YPH499 kar9Δ::klTRP1 pRS316-KAR9 GFP-Tub1-ADE2 | *Caydasi et al., 2010b* |
| AKY351 | YPH499 kar9Δ::klTRP1 pRS316-KAR9 kin4Δ::HIS3MX6 GFP-Tub1-ADE2 | *Caydasi et al., 2017* |
| AKY4001 | ESM356 leu2::LEU2-pGal1-KIN4 bud14Δ::klTRP1 | This study |
| AKY4006 | ESM356 KIN4-GFP-HIS3MX6 Spc42-eqFP-natNT2 bud14Δ::klTRP1 lte1Δ::hphNT1 mCherry-Tub1-URA3 | This study |
| AKY4007 | ESM356 Bub2-GFP-kanMX6 Spc42-eqFP-hphNT1 mCherry-Tub1-URA3 | This study |
| AKY4008 | ESM356 Tem1-GFP-kanMX6 Spc42-eqFP-hphNT1 bud14Δ::HIS3MX6 mCherry-Tub1-URA3 | This study |
| AKY4009 | ESM356 Bub2-GFP-kanMX6 Spc42-eqFP-hphNT1 bud14Δ::klTRP1 mCherry-Tub1-URA3 | This study |
| AKY4011 | ESM356 Bfa1-GFP-kanMX6 Spc42-eqFP-hphNT1 bud14Δ::klTRP1 mCherry-Tub1-URA3 | This study |
| AKY4012 | ESM356 Bfa1-GFP-kanMX6 Spc42-eqFP-hphNT1 mCherry-Tub1-URA3 | This study |
| AKY4013 | ESM356 Tem1-GFP-kanMX6 Spc42-eqFP-hphNT1 mCherry-Tub1-URA3 | This study |
| AKY4016 | ESM356 KIN4-GFP-HIS3MX6 Spc42-eqFP-natNT2 lte1Δ::URA3 mCherry-Tub1-klTRP1 | This study |
| AKY4028 | FAY145 lte1Δ::kanMX6 bud14Δ::HIS3MX6 | This study |
| AKY4036 | FAY145 lte1Δ::kanMX6 bud14Δ::HIS3MX6 pRS416-endogenous BUD14-promoter- GFP-BUD145A | This study |
| AKY4038 | FAY145 lte1Δ::kanMX6 bud14Δ::HIS3MX6 pRS416-endogenous BUD14-promoter- GFP-BUD14ΔSH3 | This study |
| AKY404 | ESM356 KIN4-GFP-HIS3MX6 Spc42-eqFP-natNT2 rts1Δ::klTRP1 | |
| AKY4040 | ESM356 Spc42-eqFP-hphNT1 BUD14-GFP-his3MX6 mCherry-Tub1-URA3 | This study |
| AKY4068 | YPH499 kar9Δ::klTRP1 pRS316-KAR9 GFP-Tub1-ADE2 bud14Δ::HIS3MX6 kin4Δ::hphNT1 | This study |
| AKY4078 | PAY704 MATa ade2-1 his3-11,15 leu2-3,112 ura3-1 can1-100 ssd1-d2 glc7::LEU2 trp1-1::GLC7::TRP1 kar9Δ::hphNT1 | This study |
| AKY4079 | PAY701 MATa ade2-1 his3-11,15 leu2-3,112 ura3-1 can1-100 ssd1-d2 glc7::LEU2 trp1-1::glc7-12::TRP1 kar9Δ::hphNT1 | This study |
| AKY4087 | ESM356 cdc15::CDC15-as1(L99G)-URA3 Bfa1-3HA- klTRP1 pGal1-Bud14-natNT2 | This study |
| AKY4088 | ESM356 Mob1-GFP-kanMX6 Spc42-eqFP-hphNT1 kin4Δ::natNT2 bud14Δ::HIS3MX6 | This study |
| AKY4091 | YPH499 cdc5-10 Bfa1-3HA-klTRP1 lte1Δ::natNT2 bud14Δ::HIS3MX6 | This study |
| AKY4094 | YPH499 cdc14-2 pRS316-CDC14 bfa1Δ::klTRP3 | This study |
| AKY4095 | YPH499 dbf2-2 pRS316-DBF2 bfa1Δ::klTRP3 | This study |
| AKY4102 | ESM356 Mob1-GFP-kanMX6 Spc42-eqFP-hphNT1 kin4Δ::natNT2 bud14Δ::HIS3MX6 mCherry-Tub1-LEU2 | This study |

*Table 1 continued on next page*

*Table 1 continued*

| Strain name | Description | Reference |
|---|---|---|
| AKY415 | ESM356 KIN4-6HA-hphNT1 rts1Δ::klTRP1 | |
| BKY032 | ESM356 BFA1-GFP-kanMX6 Spc42-eqFP-hphNT1 kar9Δ::klTRP1 pRS316-KAR9 kin4Δ::HIS3MX6 | |
| DGY001 | ESM356 KIN4-6HA-hphNT1 bud14Δ::klTRP1 | This study |
| DGY004 | ESM356 KIN4-GFP-HIS3MX6 Spc42-eqFP-natNT2 bud14Δ::klTRP1 | This study |
| DKY056 | YPH499 bud14Δ::klTRP3 | This study |
| DKY057 | YPH499 cdc14-2 pRS316-CDC14 bud14Δ::klTRP3 | This study |
| DKY058 | YPH499 mob1-67 pRS316-MOB1 bud14Δ::klTRP3 | This study |
| DKY060 | YPH499 dbf2-2 pRS316-DBF2 bud14Δ::klTRP3 | This study |
| DKY061 | YPH499 cdc15-1 pRS316-CDC15 bud14Δ::klTRP3 | This study |
| DKY063 | ESM356 KIN4-GFP-HIS3MX6 Spc42-eqFP611-natNT2 bud14Δ::klTRP1 | This study |
| DKY069 | FAY145 lte1Δ::kanMX6 pRS316-LTE1 bud14Δ::HIS3MX6 | This study |
| DKY070 | PAY704 MATa ade2-1 his3-11,15 leu2-3,112 ura3-1 can1-100 ssd1-d2 glc7::LEU2 trp1-1::GLC7::TRP1 | This study |
| DKY071 | PAY701 MATa ade2-1 his3-11,15 leu2-3,112 ura3-1 can1-100 ssd1-d2 glc7::LEU2 trp1-1::glc7-12::TRP1 | This study |
| DKY074 | PAY701 glc7::LEU2 trp1-1::glc7-12::TRP1 lte1Δ::HIS3MX6 | This study |
| DKY075 | PAY704 glc7::LEU2 trp1-1::GLC7::TRP1 lte1Δ::HIS3MX6 | This study |
| DKY078 | PAY704 glc7::LEU2 trp1-1::GLC7::TRP1 lte1Δ::HIS3MX6 pRS316-LTE1 | This study |
| DKY079 | PAY701 glc7::LEU2 trp1-1::glc7-12::TRP1 lte1Δ::HIS3MX6 pRS316-LTE1 | This study |
| DKY080 | PAY701 glc7::LEU2 trp1-1::glc7-12::TRP1 lte1Δ::HIS3MX6 pRS316-LTE1 spo12Δ::hphNT1 | This study |
| DKY081 | ESM356 BFA1-GFP-kanMX6 Spc42-eqFP hphNT1 kar9Δ::klTRP1 pRS316-KAR9 bud14Δ::HIS3MX6 | This study |
| DKY101 | ESM356 GAL1-UPL-TEM1::klTRP1 Bfa1-3HA-hphNT1 | This study |
| DKY113 | SGY37-VIII,3 bub2Δ::hphNT1 | This study |
| DKY115 | ESM356 Bfa1-3HA-hphNT1 kin4Δ::natNT2 pWS103 (pGal1-UPL-Tem1-klTRP1) | This study |
| DKY118 | ESM356 GAL1-UPL-TEM1::klTRP1 Bfa1-3HA-hphNT1 lte1Δ::natNT2 | This study |
| DKY123 | YPH499 cdc5-10 Bfa1-3HA-hphNT1 | This study |
| DKY125 | ESM356 GAL1-UPL-TEM1::klTRP1 Bfa1-3HA-hphNT1 kin4Δ::natNT2 bud14Δ::HIS3MX6 | This study |
| DKY126 | ESM356 GAL1-UPL-TEM1::klTRP1 Bfa1-3HA-hphNT1 lte1Δ::natNT2 bud14Δ::HIS3MX6 | This study |
| DKY131 | FAY145 lte1Δ::kanMX6 pRS316-LTE1 bud14Δ::HIS3MX 6GFP BUD14 CEN URA | This study |
| DKY132 | ESM356 bud14Δ::klTRP GFP BUD14 CEN URA | This study |
| DKY135 | FAY145 lte1Δ::kanMX6 bud14Δ::HIS3MX6 pRS416-ADH-BUD14 | This study |
| DKY137 | FAY145 lte1Δ::kanMX6 bud14Δ::HIS3MX6 pRS416-ADH-BUD14-F379A | This study |
| DKY145 | ESM356 kar9Δ::HIS3MX6 bud14Δ::klTRP1 pRS316 | This study |
| DKY147 | ESM356 kar9Δ::HIS3MX6 bud14Δ::klTRP1 pMK125 | This study |
| DKY149 | ESM356 kar9Δ::HIS3MX6 bud14Δ::klTRP1 pMK131 | This study |
| DKY167 | ESM356 KIN4-GFP-HIS3MX6 Spc42-eqFP-natNT2 bud14Δ::klTRP1 lte1Δ::hphNT1 | This study |
| DKY179 | YPH499 cdc5-10 pRS316-CDC5 bud14Δ::klTRP3 | This study |
| DKY188 | ESM356 KIN4-GFP-HIS3MX6 Spc42-eqFP-natNT2 lte1Δ::URA3 | This study |
| ESM2156 | YPH499 GFP-Tub1-URA3 dyn1Δ::klTRP1 kin4Δ::hphNT1 | *Pereira and Schiebel, 2005* |
| ESM2282 | YPH499 cdc15-1 pRS316-CDC15 | *Caydasi et al., 2017* |

*Table 1 continued*

| Strain name | Description | Reference |
|---|---|---|
| ESM2283 | *YPH499 dbf2-2 pRS316-DBF2* | *Caydasi et al., 2017* |
| ESM2285 | *YPH499 mob1-67 pRS316-MOB1* | *Caydasi et al., 2017* |
| ESM2326 | *ESM356 KIN4-6HA-hphNT1* | *Caydasi et al., 2010b* |
| ESM356 | *MATa ura3-52 leu2Δone his3Δ200 trp1Δ63* | *Pereira and Schiebel, 2001* |
| FAY145 | *MATa ura3-52 his3Δ200 leu2Δone lte1Δ::kanMX6 pRS316-LTE1* | *Bertazzi et al., 2011* |
| GPY1033 | *ESM356-1 KIN4-GFP-his3MX6 Spc42-eqFP-natNT2* | *Pereira and Schiebel, 2005* |
| GPY1054 | *ESM356 Mob1-GFP-kanMX6 Spc42-eqFP-hphNT1* | This study |
| GPY491 | *YPH499 cdc14-2 pRS316- CDC14* | *Caydasi et al., 2017* |
| GYBY005 | *ESM356 Glc7-TAP-kanMX (CBP-Tev-4ProtA-kanMX6)* | This study |
| HKY099 | *ESM356 BFA1-GFP-kanMX6 Spc42-eqFP-hphNT1 kar9Δ::klTRP1 pRS316-KAR9 mCherry-Tub1-LEU2* | This study |
| HKY100 | *ESM356 BFA1-GFP-kanMX6 Spc42-eqFP-hphNT1 kar9Δ::klTRP1 pRS316-KAR9 kin4Δ::HIS3MX6 mCherry-Tub1-LEU2* | This study |
| HKY101 | *ESM356 BFA1-GFP-kanMX6 Spc42-eqFP-hphNT1 kar9Δ::klTRP1 pRS316-KAR9 bud14Δ::HIS3MX6 mCherry-Tub1-LEU2* | This study |
| HKY114 | *ESM356 Bfa1-GFP-kanMX6 Spc42-eqFP-hphNT1 mCherry-Tub1-LEU2 pRS416* | This study |
| HKY115 | *ESM356 Bfa1-GFP-kanMX6 Spc42-eqFP-hphNT1 bud14Δ::klTRP1 mCherry-Tub1-LEU2 pRS416* | This study |
| HKY116 | *ESM356 Bfa1-GFP-kanMX6 Spc42-eqFP-hphNT1 bud14Δ::klTRP1 mCherry-Tub1-LEU2 pRS416-BUD14* | This study |
| HKY133 | *FAY145 lte1Δ::kanMX6 pRS316-LTE1 kel1Δ::hphNT1* | This study |
| HKY134 | *FAY145 lte1Δ::kanMX6 pRS316-LTE1 kel2Δ::hphNT1* | This study |
| HKY135 | *FAY145 lte1Δ::kanMX6 pRS316-LTE1 spo12Δ::natNT2 kel1Δ::hphNT1* | This study |
| HKY136 | *FAY145 lte1Δ::kanMX6 pRS316-LTE1 spo12Δ::natNT2 kel2Δ::hphNT1* | This study |
| HKY139 | *ESM356 kar9Δ::HIS3MX6 pRS316-KAR9 kel1Δ::hphNT1* | This study |
| HKY140 | *ESM356 kar9Δ::HIS3MX6 pRS316-KAR9 kel2Δ::hphNT1* | This study |
| HKY155 | *ESM356 BUD14-TAP-kanMX6* | This study |
| HKY164 | *ESM356 Bud14-TAP-kanMX6 Glc7-9myc-HIS3MX6* | This study |
| HKY171 | *ESM356 Mob1-GFP-kanMX6 Spc42-eqFP-hphNT1 mCherry-Tub1-LEU2* | This study |
| HKY172 | *ESM356 Mob1-GFP-kanMX6 Spc42-eqFP-hphNT1 mCherry-Tub1-LEU2 bud14Δ::HIS3MX6* | This study |
| HKY173 | *ESM356 Mob1-GFP-kanMX6 Spc42-eqFP-hphNT1 mCherry-Tub1-LEU2 kin4Δ::natNT2* | This study |
| HKY174 | *ESM356 Mob1-GFP-kanMX6 Spc42-eqFP-hphNT1 mCherry-Tub1-LEU2 bfa1Δ::natNT2* | This study |
| HKY175 | *YPH499 kar9Δ::klTRP1 pRS316-KAR9 GFP-Tub1-ADE2 bud14Δ::HIS3MX6 bfa1Δ::natNT2* | This study |
| HKY177 | *YPH499 bfa1Δ::klTRP1* | This study |
| HKY180 | *YPH499 mob1-67 pRS316-MOB1 bfa1Δ::klTRP3* | This study |
| HKY182 | *YPH499 cdc15-1 pRS316-CDC15 bfa1Δ::klTRP3* | This study |
| HKY183 | *YPH499 cdc5-10 pRS316-CDC5 bfa1Δ::klTRP3* | This study |
| HKY184 | *ESM356 cdc15::CDC15-as1(L99G)-URA3 Bfa1-3HA- klTRP1 bud14Δ::HIS3MX6* | This study |
| HKY185 | *ESM356 pRS416* | This study |
| HKY186 | *ESM356 pMK60 (Bud14-GFP)* | This study |
| HKY187 | *ESM356 pHK002 (Bud14-F379A-GFP)* | This study |

*Table 1 continued on next page*

*Table 1 continued*

| Strain name | Description | Reference |
| --- | --- | --- |
| HKY188 | *ESM356 pDK003 (Bud14-ΔSH3-GFP)* | This study |
| HKY189 | *ESM356 pDK001 (Bud14-5A-GFP)* | This study |
| HKY190 | *ESM356 BFA1-GFP-kanMX6 Spc42-eqFP-hphNT1 bud14Δ::klTRP1 mCherry-TUB1-Leu2 pMK131* | This study |
| IKY075 | *SGY37-VIII,3 kin4Δ::klTRP1* | This study |
| DKY133 | *SGY37-VIII,3 bub2Δ::hphT1* | This study |
| IKY192 | *ESM356 kar9Δ::HIS3MX6 bud14Δ::klTRP1 pRS416-endogenous BUD14-promoter- GFP-BUD14-5A* | This study |
| IKY193 | *ESM356 kar9Δ::HIS3MX6 bud14Δ::klTRP1 pRS416-endogenous BUD14-promoter- GFP-BUD14-ΔSH3* | This study |
| JOY79 | *YPH499 cdc5-10 pRS316-CDC5* | *Caydasi et al., 2017* |
| PAY701 | *W303 MATa ade2-1 his3-11,15 leu2-3,112 ura3-1 can1-100 ssd1-d2 glc7::LEU2 trp1-1::glc7-12::TRP1* | *Andrews and Stark, 2000* |
| PAY704 | *W303 MATa ade2-1 his3-11,15 leu2-3,112 ura3-1 can1-100 ssd1-d2 glc7::LEU2 trp1-1::GLC7::TRP1* | *Andrews and Stark, 2000* |
| SEY032 | *YPH499 kar9Δ::klTRP3 pRS316-KAR9 GFP-Tub1-ADE2 bud14Δ::HIS3MX6* | This study |
| SEY206 | *ESM356 cdc15::CDC15-as1(L99G)-URA3 Bfa1-3HA- klTRP1* | This study |
| SEY212 | *ESM356 cdc15::CDC15-as1(L99G)-URA3 Bfa1-3HA- klTRP1 Bud14-9myc-HIS3MX6* | This study |
| SEY218 | *ESM356 cdc15::CDC15-as1(L99G)-URA3 Bfa1-3HA- klTRP1 bud14Δ::HIS3MX6 leu2::LEU2- pGal1-Bud14-9myc-hphNT1* | This study |
| SEY219 | *ESM356 cdc15::CDC15-as1(L99G)-URA3 Bfa1-3HA- klTRP1 bud14Δ::HIS3MX6 leu2::LEU2- pGal1-Bud14- F379A-9myc-hphNT1* | This study |
| SGY37-VIII,3 | *MATa leu2 his3 trp1 ADE2 ura3-52::URA3-lexA-op-LacZ* | *Geissler et al., 1996* |
| SHM562 | *YPH499 GFP-Tub1-URA3 dyn1Δ::klTRP1* | *Maekawa et al., 2007* |
| YDA101 | *ESM356 kar9Δ::HIS3MX6 bud14Δ::klTRP1* | *Caydasi and Pereira, 2017* |
| YPH499 | *MATa ura3-52 lys2-801 ade2-101 trp1Δ63 his3Δ200 leu2Δ1* | *Sikorski and Hieter, 1989* |

Then cells were released from α-factor in YPAD. The arrest was confirmed by microscopy after ethanol fixation of the cells and staining the nucleus with DAPI. As for anaphase arrest using *cdc15-as*, log-phase cell cultures grown in YP-raffinose medium were treated with 1NM-PP1 (2.5 µM) and incubated for 3 hr. Synchronization was judged based on microscopy after DAPI staining. Anaphase synchronization of *cdc5-10* cells was achieved by arresting the cells in G1 at 23°C first and then releasing them in prewarmed alpha factor-free medium at 30°C and incubating them at 30°C for 1.5 hr.

## Microscopy methods

All fluorescence microscopy methods were performed using Carl Zeiss Axio Observer 7 motorized inverted epifluorescence microscope equipped with Colibri 7 LED light source, Axiocam 702 Mono-chrome camera, 100× and 63× Plan Apochromat immersion oil objective lenses, Filter sets 95 (GFP/Cherry), 20 (Rhodamin), FITC and DAPI, and an XL incubation and climate chamber. For Kin4-GFP, GFP-tubulin and Bud14-GFP analysis 63× objective was used. For analysis of Bfa1-GFP, Bub2-GFP, Tem1-GFP, and Mob1-GFP 100× objective was used.

For endpoint analysis of samples, wet mounts of samples were prepared by sandwiching 2 µl of culture between the coverslip and slide. Cells were either fixed using 70% ethanol and stained with DAPI (for SPOC assays and mitotic index analysis) or analyzed live (for GFP signal quantification). For time-lapse experiments, cells were prepared on glass-bottom Petri dishes (WVR 10810-054 Matsunami) as described (*Caydasi and Pereira, 2017*). Briefly, the glass center of the Petri dish was covered with 6% Concanavalin A (*Canavalia ensiformis* Jack Bean, type IV Sigma C2010-G). Excess

**Table 2.** Plasmids used in this study.

| Plasmid name | Description | Reference |
|---|---|---|
| pRS316 | *URA3-dependent CEN-based yeast-E. coli* shuffle plasmid *AmpR* | *Sikorski and Hieter, 1989* |
| pRS416 | *URA3-dependent CEN-based yeast-E. coli* shuffle plasmid *AmpR* | *Geissler et al., 1996* |
| pAFS125 | *GFP-TUB1-URA3*-containing integration plasmid | *Straight et al., 1997* |
| pSM1027 | *GFP-TUB1-ADE2*-containing integration plasmid | *Caydasi et al., 2010b* |
| pAK010 | *mCherry-TUB1-kιTRP1*-containing integration plasmid | *Khmelinskii et al., 2007* |
| pAK011 | *mCherry-TUB1-URA3*-containing integration plasmid | *Khmelinskii et al., 2007* |
| pBK067 | *mCherry-TUB1-LEU2*-containing integration plasmid | *Caydasi et al., 2014* |
| pWS103-1 | *pRS304-Gal1-UPL-TEM1* | *Shou et al., 1999* |
| pMM5 | p423-Gal1-lexA-Myc | *Geissler et al., 1996* |
| pMM6 | p425-Gal1-Gal4-HA. | *Geissler et al., 1996* |
| pCDV471 | *pEG202 ADH1-LexA-DBD-GLC7* | *Lenssen et al., 2005* |
| pEG202 | *2 µm ADH1-LexA-DBD-HIS3* | *Pereira et al., 1999* |
| pCL2-1 | *pMM6-BUB2* | *Höfken and Schiebel, 2004* |
| pCL4a-3 | *pMM6-BFA1* | *Höfken and Schiebel, 2004* |
| pHA132-2 | *pMM6-KIN4 (1–750 aa)* | Gift from G. Pereira |
| pHA69-1 | *pMM5-BUD14* | Gift from G. Pereira |
| pHA70-1 | *pMM6-BUD14* | Gift from G. Pereira |
| pTH221 | *pMM6-KEL* | *Höfken and Schiebel, 2004* |
| IKY006 | *pMM5-BUD14-5A* | This Study |
| IKY007 | *pMM5-BUD14-F379A* | This Study |
| IKY008 | *pMM5-BUD14-ΔSH3* | This Study |
| IKY010 | *pMM6-BUD14-5A* | This Study |
| IKY011 | *pMM6-BUD14-F379A* | This Study |
| IKY012 | *pMM6-BUD14-ΔSH3* | This Study |
| pMK125 | *pRS416 ADH-BUD14* | *Knaus et al., 2005* |
| pMK131 | *pRS416 ADH-BUD14-F379A* | *Knaus et al., 2005* |
| pMK60 | *pRS416-endogenous BUD14-promoter-GFP-BUD14* | *Knaus et al., 2005* |
| pDKY001 | *pRS416-endogenous BUD14-promoter-GFP-BUD14$^{5A\ (135A\ 137A\ 138A\ 139A\ 140A)}$* | This Study |
| pDKY003 | *pRS416-endogenous BUD14-promoter-GFP-BUD14$^{ΔSH3(Δaa\ 262–318)}$* | This Study |
| pHK002 | *pRS416-endogenous BUD14-promoter-GFP-Bud14-F379A* | This study |
| pGW399-1 | *pRS316-KAR9* | *Caydasi et al., 2010b* |
| pSM805 | *pRS305-pGal1* | *König et al., 2010* |
| pHK003 | *pSM805 pGal1-Bud14-Leu2* | This study |
| pHK004 | *pSM805 pGal1-Bud14-F379A-Leu2* | This study |
| pIK007 | *pMM5-Bud14-F379A* | This study |
| pSM903-4 | *pRS316-LTE1* | *Höfken and Schiebel, 2002* |
| pSM908 | *pRS316-DBF2* | *Caydasi et al., 2017* |
| pSM926 | *pRS316-MOB1* | *Caydasi et al., 2017* |
| pUG120 | *pRS316-CDC14* | *Gruneberg et al., 2000* |

*Table 2 continued on next page*

*Table 2 continued*

| Plasmid name | Description | Reference |
|---|---|---|
| pBS9 | *pRS316-CDC15* | *Caydasi et al., 2017* |
| pCL33 | *pRS316-CDC5* | *Höfken and Schiebel, 2004* |

Concanavalin A was washed out and 150 µl of culture was pipetted on the dish followed by a 30 min incubation at 30°C. Then, cells were aspirated gently and cells that were not attached on the dish were washed out with prewarmed fresh media. After the final wash, the dish was filled with media and taken to the microscope stage in the preheated chamber, equilibrated there for at least 1 hr before the start of the time lapse.

## SPOC functionality assays

For endpoint analysis of SPOC functionality, log-phase *kar9Δ* cells cultured at 23°C were incubated at 30°C for 3–5 hr or log-phase *dyn1Δ* cells cultured at 30°C were incubated at 18°C overnight. Cells were fixed using 70% ethanol, resuspended in PBS containing 1 µg/ml DAPI, and analyzed by microscopy. Cells bearing normal and misaligned nuclei, and cells with SPOC-deficient phenotypes including multinucleated cells were counted. SPOC deficiency index was calculated as

$$\textit{Cells with multinucleation} \div \textit{cells with mispositioned nuclei} \times 10$$

For time-lapse analysis of SPOC functionality, GFP-tubulin-bearing *kar9Δ* cells were imaged every minute for 90 min at 30°C. 10 z-stacks of 0.4 µm thickness were taken at each time point. Anaphase duration of cells with correct and misaligned spindles was calculated as the time from the start of fast spindle elongation (metaphase-to-anaphase transition) until spindle breakdown (*Caydasi et al., 2010b*; *Straight et al., 1997*). Dotplots were prepared in GraphPad Prism 8.0.1 (GraphPad, La Jolla, CA).

## Localization-based fluorescence image quantifications

For signal intensity quantification of Bfa1-GFP, Bub2-GFP, Tem1-GFP, and Mob1-GFP, we took still images of live cells bearing *SPC42*-eqFP and mCherry-*TUB1* as a mitotic marker. Each still image consisted of 13 z-stacks of 0.3 µm thickness. Z-stacks of the images were sum-projected, and mean fluorescence intensities were measured using ImageJ (NIH) software and corrected as described (*Caydasi et al., 2012*). Briefly, a 0.577 µm² (42 pixels) box was selected around the SPB under inspection (region of interest [ROI]) and the mean fluorescence intensity of the GFP signal was measured. Background fluorescence intensity was determined by measuring an area inside the cytoplasm of respective cell, near to the ROI. The correction was done by subtracting the mean fluorescence intensity of the background from the mean fluorescence intensity of the ROI.

To detect the behavior of asymmetric localization of Bfa1 at SPBs, ratios of the corrected signal intensities (as described above) at the SPBs were calculated. For cells with normally aligned spindles, the signal intensity at the dSPB was divided to the signal intensity at the mSPB. For cells with misaligned spindles, the SPB with the greater signal was considered as dSPB.

Statistical results and graphs were computed and plotted by using Prism 8.0.1 (GraphPad). Two-tailed t-tests or one-way anova were applied to compare the statistical significance of the signal intensity measurements in a pairwise manner.

## Yeast two hybrid assay

Yeast two hybrid analysis was performed as described (*Meitinger et al., 2014*). Briefly, indicated genes cloned into pMM5 or pEG202 (LexA-myc fusion, *HIS* autotrophy marker) and pMM6 (Gal4-HA fusion, *LEU2* autotrophy marker) were transformed into the yeast strain SGY37-VIII,3 that contains a chromosomally integrated LexA(op)-LacZ reporter for activation of the ß-galactosidase gene. Transformants were selected on SC-His-Leu agar plates, grown in SC-His-Leu media until stationary phase, diluted to 1 OD$_{600}$ and 10 µl of each were spotted on SC-His-Leu agar plates. Cells were grown

for ~48 hr, and plates were overlaid with 0.4% soft-agar solution containing 0.4 mg/ml X-Gal, 0.1% SDS, 10 mM KCl, and 1 mM $MgCl_2$. Blue color formation was monitored. Plates were scanned.

## Glc7 functionality assay

For the analysis of *GLC7* functionality, *GLC7* and *glc7-12* cells were grown at 23°C until log-phase. Log-phase cultures were diluted to a $OD_{600}$ value of 0.1 and shifted to 30, 33, 35, and 37°C. After incubation at these temperature for 4 hr, cells were fixed using 70% ethanol, and then resuspended in with 1 µg/ml DAPI in 1× PBS prior to microscopic visualization. Then, the cells were analyzed under fluorescence microscopy. All the cells were counted and categorized according to their bud size and nucleus number. For analyzing the functionality of *GLC7* by growth assay, the cultures were grown at 23°C for 24–48 hr. $OD_{600}$ value of all the yeast cultures, then, was adjusted to 1.0 using 1× sterile PBS. The cultures were serially diluted 10-fold using sterile 1× PBS. 7.5 µl of each serial dilutions were spotted on the appropriate agar plates (YPD and YPD-containing 0.01 M HU). The plates were incubated at the appropriate temperatures for 2–3 days.

## Protein methods

Total yeast protein precipitation, sample preparation, and western blotting following Laemmli SDS-PAGE were performed as previously described (*Meitinger et al., 2016*). Total cell native protein extracts were prepared by lysis of cells using acid-washed glass beads (Sigma) in a cooling bead beater homogenizer (Analytik Jena, SpeedMill Plus). Primary antibodies were mouse-anti-HA (gift from Gislene Pereira), rabbit anti-Clb2 (gift from Gislene Pereira), rabbit-anti-tubulin (Abcam EPR13799), rabbit anti-TAP (Thermo Fisher Scientific CAB1001), and rabbit anti-GFP (Abcam ab290). Secondary antibodies were goat-anti-rabbit (Advansta #R-05062-500) or goat-anti-mouse (Advansta #R-05071-500) horse radish peroxidase (HRP) conjugated secondary antibodies.

## Phosphatase assay

Glc7-TAP and Bud14-TAP were enriched from yeast lysates on IgG Sepharose beads (GE Healthcare BioScience) according to the manufacturer's recommendations. In addition, ESM356-1 (a strain without TAP tag) lysates were incubated with IgG Sepharose beads as a no-tag control. Briefly, cell pellets of logarithmically growing cultures were lysed using acid-washed glass beads (Sigma) in a lysis buffer containing 50 mM HEPES pH 7.5, 150 mM NaCl, 1 mM EDTA, 1% SDS, 1 mM DTT, 2 mM PMSF, and cOmplete Protease Inhibitor Cocktail (Roche). The lysate was cleared by centrifugation at 10,000 rpm at +4°C and the supernatant was incubated with IgG beads for 2 hr at 4°C. Beads were washed with ice-cold wash buffer containing 50 mM HEPES pH 7.5, 150 mM NaCl, 0.1 mM EDTA, 0.1% SDS, 0.025% Tween 20, 1 mM DTT, and were immediately used for the phosphatase reaction as the phosphatase source.

Bfa1-3HA was purified from *BFA1-3HA kin4Δ Gal1 -UPL-TEM1* cells arrested in anaphase. For this, log-phase culture grown in raffinose/galactose-containing rich medium were transferred into YPAD medium. After >90% of the cell arrest was achieved, cell pellets were lysed using acid-washed glass beads in bead beater. Lysis buffer composition was 50 mM HEPES pH 7.5, 150 mM NaCl, 5% glycerol, 1 mM EDTA, 1% NP-40, 160 mM β-glycerophosphate, 2 mM $NaVO_3$, 100 mM NaF, 2 mM PMSF, 4 mM benzamidine and cOmplete Protease Inhibitor Cocktail (Roche). After clearing the cell lysate by centrifugation at 10,000 rpm at +4°C , the cell extract was incubated with anti-HA magnetic beads (Thermo Fisher, Pierce) for 2 hr at 4°C. After the incubation, the beads were washed in 50 mM HEPES pH 7.5, 150 mM NaCl, 1% NP-40, 5% glycerol, 1 mM EDTA, and eluted from beads under basic elution conditions according to the manufacturer's protocol. Briefly, beads were incubated with 50 mM NaOH for 5 min and the eluate was immediately neutralized by 300 mM Tris pH 8.5. Next, the eluate was applied on PD MiniTrap G-25 Sephadex (GE Healthcare) columns equilibrated with a buffer containing 50 mM HEPES pH 7.5, 100 mM NaCl. Buffer exchange was performed according to the manufacturer's spin protocol.

In vitro phosphatase reaction was performed using the IgG beads that contained Bud14-TAP or Glc7-TAP or the IgG beads that was incubated with the cell extract of the ESM356-1 control. Bfa1-3HA that was prepared as explained above was added as the substrate. Phosphatase reaction buffer composition was 100 mM NaCl, 50 mM HEPES pH 7.5, 2 mM DTT, 0.025% Tween, 2 mM $MnCl_2$. Okadaic acid (Abcam) was also added to the indicated reactions to 1.5 µM final concentration. After

reaction was incubated at 30°C for 1 hr, 5× sample buffer was added immediately into the reaction tubes, then samples were boiled at 95°C for 5 min and analyzed by western blotting.

## Quantification of hyperphosphorylation of Bfa1

Bfa1-3HA-containing protein samples were run 8% Laemmli SDS-PAGE gels for about 3–3.5 hr. The gel tank was placed in an ice-filled bucked during running. Gels were run at 50 V for about 30 min (to allow passage of proteins from stacking to separating gel), and at 100 V for the remaining time. Semi-dry blotting was performed for 2 hr at 0.1 A per gel, and signals were detected on x-ray films using ECL. X-ray films were scanned, and the images were analyzed in ImageJ. Mean gray value of the top half and bottom half of the Bfa1-3HA bands at each line was quantified using ImageJ measure tool. The area sizes were kept constant for each blot. Background mean intensity of the film was also quantified and subtracted from that of the measured Bfa1-3HA band intensities for correction. Corrected intensity of the top half was divided to the corrected intensity of the bottom half of the Bfa1-3HA bands and the values were plotted.

## Acknowledgements

This work was supported by TÜBITAK grant no. 117C041. Work of AKC was supported by EMBO installation grant no. 3918 and ICGEB installation grant no. CRP/TUR17-04_EC. CD was funded by Koç University, HK and IK were funded by TÜBITAK grant no. 118Z979, and SNB was funded by TÜBITAK grant no. 117Z232 granted to AKC. AKC was funded by MSCA Individual Fellowship No. 796599-COHEMEX. We are grateful to Gislene Pereira (COS, Heidelberg), Elmar Schiebel (ZMBH, Heidelberg), Claudio De Virgilio (University of Fribourg, Fribourg), Matthias Peter (ETH, Zürich), and Michael Knop (ZMBH, Heidelberg) for sharing reagents, strains, and plasmids. We would also like to thank to Nesrin Özeren (Boğaziçi University, Istanbul), Ilke Suder (Boğaziçi University, Istanbul) and the Vivarium of Boğaziçi University for their endless support on antibody tests.

## Additional information

### Funding

| Funder | Grant reference number | Author |
| --- | --- | --- |
| EMBO | IG - 3918 | Dilara Kocakaplan<br>Ayse Koca Caydasi |
| TUBITAK | 117C041 | Ayse Koca Caydasi |
| TUBITAK | 118Z979 | Hüseyin Karaburk<br>Idil Kirdök<br>Ayse Koca Caydasi |
| European Commission | H2020-MSCA-IF (796599) | Ayse Koca Caydasi |
| ICGEB | IG-No. CRP/TUR17-04_EC | Ayse Koca Caydasi |
| TUBITAK | 117Z232 | Seyma Nur Bektas<br>Ayse Koca Caydasi |

The funders had no role in study design, data collection and interpretation, or the decision to submit the work for publication.

### Author contributions

Dilara Kocakaplan, Data curation, Formal analysis, Investigation, Methodology, Visualization; Hüseyin Karabürk, Data curation, Formal analysis, Investigation, Methodology, Validation, Visualization; Cansu Dilege, Data curation, Investigation, Methodology, Visualization; Idil Kirdök, Seyma Nur Bektas, Data curation, Investigation, Methodology, Validation, Visualization; Ayse Koca Caydasi, Conceptualization, Funding acquisition, Project administration, Supervision, Writing - original draft, Writing - review and editing

**Author ORCIDs**
Ayse Koca Caydasi http://orcid.org/0000-0003-2570-1367

**Decision letter and Author response**
Decision letter https://doi.org/10.7554/72833.sa1
Author response https://doi.org/10.7554/72833.sa2

## Additional files

**Supplementary files**
• Transparent reporting form

**Data availability**
All data generated or analysed during this study are included in the manuscript and supporting file. Source Data files have been provided for Figures 2, 3, 4, 6, 7 and Figure supplements for Figures 2, 5, 6 and 7. These include numerical values and statistics for data shown in graphs as well as labelled uncropped blot images and raw scans of the x-ray films of the immunoblots.

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
