## [Decision Letter]

**Acceptance summary:**

Spindle positioning checkpoint (SPOC) is a mechanism that prevents aneuploidy in yeast. Previous work has revealed the importance of the MEN pathway and its components in the regulation of aspects of the SPOC. Here through the use of a comprehensive and extensive array of approaches the authors propose an overlapping role for the kelch repeat protein BUD14 and protein phosphatase 1 (Glc7) in regulating SPOC. Given the role of phosphatases in spindle orientation and positioning in others organisms (such as *Drosophila*), these mechanisms may be more widespread.

**Decision letter after peer review:**

[Editors’ note: the authors submitted for reconsideration following the decision after peer review. What follows is the decision letter after the first round of review.]

Thank you for submitting your work entitled "Protein Phosphatase 1 in association with Bud14 inhibits mitotic exit in *Saccharomyces cerevisiae*" for consideration by eLife. Your article has been reviewed by 2 peer reviewers, and the evaluation has been overseen by a Reviewing Editor (Mohan Balasubramanian) and a Senior Editor. The following individual involved in review of your submission has agreed to reveal their identity: Rosella Visintin (Reviewer #2).

We are sorry to say that, after consultation with the reviewers, we have decided that your work will not be considered further for publication by eLife at present. This is largely due to the extensive amount of revisions suggested. However, given that the editors' and referees' views on the significance of the work and the strength of conclusions, we will be happy to consider a thoroughly reworked manuscript. The revisions required includes a full rewrite (Dr. Visintin has kindly provided a large number of suggestions to this effect) as well as better quantitation and a few other mechanistic experiments (suggested by both referees).

The editors and the referees are enthusiastic about this field of work and the fact that you have identified Bud14 as a novel component of the Spindle position checkpoint. They appreciate that insights developed from your work can help understand related processes in other organisms as well. The referees also appreciate the case you have built on the interplay between Bud14, PP1, and the MEN signaling network. But both referees have raised a number of major concerns that are listed verbatim below, addressing which will help in revising this work.

*Reviewer #1:*

In this study, the authors aim to uncover novel regulators of the Spindle position checkpoint (SPOC), and the results have implications in understanding the signaling cross-talk between mitotic exit and spindle position. The authors initially show that Bud14 deletion can rescue growth defects of mitotic exit deficient cells and based on these observations, they hypothesize that Bud14 may play a role in the activation of Spindle Position Checkpoint (SPOC). They use genetic, microscopic, and biochemical tools to test their hypothesis. The genetic and microscopic analysis presented in the manuscript demonstrates that Bud14 is involved is directly involved in SPOC activation upon spindle misalignment and is convincing despite a few minor issues. They also provide data that this effect exerted by Bud14 on SPOC may require its interaction with PP1 phosphatase Glc7; however, the involvement of Glc7 is not directly shown in all assays. Building on the genetic evidence, the biochemical analysis presented in the manuscript attempts to test whether Glc7 in association with Bud14 dephosphorylates Bfa1 and helps in its activation, but the biochemical data shown to support this claim is not sufficiently convincing and has problems in experimental design and interpretation. The data does not provide clear evidence to demonstrate that Glc7 dephosphorylates Bfa1 and lacks empirical evidence of the effect of the same on Bfa1. Overall, the study succeeds in building a preliminary case for the role of Glc7-Bud14 but lacks the mechanistic details of how they regulate bfa1 activity during normal or misaligned spindle positioning.

In this manuscript Kocakaplan et al, have identifies the role of Bud14-Glc7 as a novel regulator of SPOC by controlling Bfa1 activity through dephosphorylation. This study provides a very preliminary insight into Bud14-Glc7 role in Spindle Position Checkpoint. In addition, the majority of the results presented in this manuscript need further detailed experiments and analysis to support the claims made by the authors. Some of the interpretations of data are largely not convincing (i.e., Glc7-Bud14 dependent dephosphorylation of Bfa1 in vivo and in vitro). Further mechanistic insights of Glc7-Bud14 role in regulating MEN component (Bfa1) are strongly required. Over all the initial findings by Kocakaplan et al seem interesting observations in the field of SPOC and MEN regulation.

1) In Figure 1, the authors have shown that Bud14 genetically interacts with MEN components and Bud14 deletion rescues growth defects of MEN-ts mutants. Based on these observations, they hypothesize that Bud14 may be involved in mitotic exit inhibition and acting upstream of MEN pathway. The data presented support the conclusion, but the authors should mention the restrictive temperatures of MEN-ts mutants used in Figure 1C.

2) In Figure 2, the authors show that Bud14 deletion causes a deficiency of SPOC arrest upon spindle misalignment by comparing anaphase duration with Kar9 null cells. Bud14 is also shown to have synthetic sick interaction with Kar9. The behaviour of Bud14 is similar to the behaviour of Kin4, which is already known to be involved in SPOC activation. The data presented in the figure collectively suggest a role for Bud14 in SPOC arrest activation and supports the conclusions made by the authors. However, there are no statistics shown in Figure 2A and 2B to determine whether the differences observed are significant or not.

3) In Figure 3, the authors show that Glc7-Bud14 interaction is important for Bud14 role in SPOC and its function in SPOC is independent of its function in Bnr1-regulation. The data supports the conclusions well except for the following discrepancies:

a) In Figure 3B, glc7 only rescues the lethality of lte1Δspo12Δ cells at its semi-permissive temperature (33C). This raises the question whether this condition represents a Glc7 deficient condition. The authors should address this.

b) No statistics in Fig 3C to ascertain significance between differences.

c) In Figure 3D, the introduction of bud14-F379A mutant in kar9Δbud14Δ cells shows partial rescue of the multi-nucleate phenotype of kar9Δbud14Δ cells (compare black bars). This observation challenges the importance of Glc7-Bud14 interaction in activating SPOC. Also, no statistics is provided to ascertain the significance of differences.

4) In Figure 4, the authors demonstrate that Bud14 interacts with Bfa1 and Bub2 is important for this interaction. The authors should also check these interactions using the already used Bud14 mutants in Figure 3 to get more insight into the nature of these interactions. In addition to Bud14, they should also check Glc7 interaction with these proteins to support the idea that Glc7-Bud14 interaction is necessary for SPOC activation.

5) In Figure 5, the authors show that more Bfa1 associates with SPBs, especially dSPB in Bud14Δ cells. They also show that unlike Kin4 deletion, Bud14 depletion does not change in the asymmetric localization of Bfa1 between the two SPBs but merely causes an increase in Bfa1 accumulation at SPBs irrespective of spindle misalignment. These observations suggest a role for Bud14 in controlling Bfa1 sequestration at the SPB and further support the idea that Kin4 and Bud14 work independently in different pathways in SPOC.

6) In Figure 6, the authors show that Bfa1 may be dephosphorylated by Bud14 as there is the accumulation of hyper-phosphorylated Bfa1 in lte1Δbud14Δ and lte1Δkin4Δ cells. The authors argue that like Kin4, Bud14 activity may be counteracting the effect of Cdc5 kinase-mediated phosphorylation of Bfa1. However, the data has the following discrepancies:

7) In Figure 6C, the effect of the phosphatase inhibitor Okadaic acid is not striking even at high concentration (lane 3). Since the effect is so small, I suggest the authors to quantify the effect for better visualization of the reader.

8) Based on Figure 6C, the authors make the conclusion that Glc7 promotes dephosphorylation of Bfa1 (lines 342-343 in text), but Okadaic acid is not a specific inhibitor for Glc7. It can also inhibit Cdc55-PP2A and other phosphatases (i.e., Cdc14), which is known to dephosphorylate Bfa1 so this conclusion cannot be made until the contribution of Cdc55 can be separated from the contribution of Glc7.

9) In Figure 6D, the bands in Lane 1,2,3 show very little presence of Bfa1-HA (measured using anti-HA antibody) as compared to Lanes 4,5,6. Considering that equal amounts of Bfa1-HA were used for the in vitro phosphatase assay, the levels of Bfa1-HA in all lanes should be approximately equal. This can potentially confound analysis of the assay. The authors should also check Bfa1 dephosphorylation in the presence of both Bud14 and Glc7 as that represents the in vivo condition. The experiment should be replicated to prove the reproducibility of the observed result.

10) In Figure 6D, Bud14 addition has no visible effect on the phosphorylation status of Bfa1, but the authors come to a totally opposite conclusion in Lines 349-352 in the text.

11) In Figure 7, the authors show that the percentage of cells with Bud14 showing bud cortex localization decreases upon spindle misalignment, but this could just be a consequence of prolonged mitotic arrest due to SPOC. The data presented in the figure does not contribute much to the main story of the paper. As an addition, the authors should also check co-localization of Bud14, Glc7 and Bfa1 during the cell cycle in cells with normal and misaligned spindles to test their hypothesis that Glc7-Bud14 complex dephosphorylates Bfa1 to activate SPOC.

In summary, the manuscript needs major revision with more emphasis on uncovering the molecular mechanism of how Glc7-Bud14 action on Bfa1 helps in SPOC.

*Reviewer #2:*

In the manuscript "Protein Phosphatase 1 in association with Bud14 inhibits mitotic exit in *Saccharomyces cerevisiae*" Kocakaplan and colleagues identify Bud14 as a novel component of the Spindle Positioning Checkpoint (SPoC) in budding yeast. The SPoC is a surveillance pathway that prevents mitotic exit until the correct spindle position is achieved, hence preserving genome integrity. Importantly, this mechanism is likely to be conserved, at least functionally, in higher eukaryotes, making its characterization of high relevance. Although Bud14 was previously linked to the regulation of spindle positioning - through the modulation of Dyn1 and actin cables - this work unveils a novel connection between Bud14 and the regulation of mitotic exit. The authors show that Bud14 in complex with the PP1 phosphatase Glc7 de-phosphorylates the checkpoint effector Bfa1, thereby fully activating it.

The authors succeeded in the characterization of Bud14 as a novel component of the checkpoint; in the identification of its role within the pathway, that is via regulation of type 1 phosphatase Glc7 and in identifying Bfa1 as a target for the Bud14-Glc7 complex. The majority of the data supporting these aspects are solid, well controlled and sustained by the appropriate statistical analyses. Statistics for Figure 2C could be worked out better.

The paper is weaker in respect to the characterization of the molecular mechanism by which the Bud14-Glc7 complex affects Bfa1 activity. For this section data need to be clarified and extended.

Although I found the topic interesting and most of the data solid, I found the manuscript very difficult to read. My major criticism of the manuscript concerns the logic of the storyline. In particular it was not always clear to me the rationale by which experiments were performed and interpreted and consequently I found confusing the sequence in which the data were presented. I believe that the manuscript will benefit from a substantial re-writing and re-organization of the data. For the above-mentioned reason I organized my review into two distinct sections: the first aimed at suggesting an alternative way of presenting the data, the second focused on addressing few experimental concerns, whose elucidation could contribute in strengthening the authors conclusions.

Part one:

Following the identification of Bud14 as a novel mitotic exit inhibitor, the authors move to investigate its potential role in the SPoC. I would re-organize the manuscript starting from here:

Q1: Is Bud14 a component of the SPoC?

Rationale1: Inactivation of SPoC in cells with mis-aligned spindles results in formation of multinucleate and anucleate cells.

Experimental design1: assess the consequences of deleting BUD14 in cells lacking Kar9 or Dyn1 - both presenting mis-aligned spindles (I would introduce Dyn1 with the same logic used for Kar9) - looking at multinucleate cells as output

Results1: In both cases the percentage of multinucleate cells increases indicating that bud14Δ cells are defective in SPoC

Rationale1.2: To further investigate the role of Bud14 in SPoC assess the anaphase residence timing

Results1.2: Bud14 is a bona fide SPoC component

Q2: Is Bud14 working together or parallel to Kin4?

Rationale2: Epistasis experiments: kin4 bud14 double mutant and GAL-KIN4 bud14

Results2: They work in parallel

Q3: Which function of Bud14 is required for its SpoC activity?

Rationale3: Bud14 is reported to act as an inhibitor of formin and a regulatory subunit of PP1 Glc7. Assess whether one of these functions is required for SPoC activity

Experimental design3.1: Formin/actin experiment

Results3.1: Bud14 does not contribute to the SpoC via formin or actin regulation

Experimental design3.2: Test for Glc7 function

Results3.1: Bud14 contributes to SPoC activity at least in part through its association with Glc7. However the obs that lack of the SH3 domain leads to a more severe defect calls for an additional mechanism, likely involving Bud14 cortex localization

Q4: What is Bud14-Glc7 target within the SPoC?

Rationale4: the obs that: i)Kin4 targets Bfa1; ii)the multinucleate defect of kin4bud14 double mutants in kar9delta cells resembles the one of kar9bfa1 double mutants identify Bfa1 as a putative candidate

Experimental design4: 2-hybrid experiment and phosphorylation/de-phosphorylation experiments

Results4: Bfa1 is a target of the Bud14-Glc7 complex

Q5: How does Bud14-Glc7 affect Bfa1 activity?

Rationale5.1: Bfa1 regulated by changes in subcellular localization

Experimental design5.1: Test Bfa1 localization in cells lacking Bud14

Results5.1: does not seem to affect localization but rather Bfa1 amount

Q6: Does Bud14-Glc7 counteract Cdc5-mediated Bfa1 inhibitory phopshorylation?

Rationale5.2: Bfa1 is inhibited by Cdc5

Experimental design5.2: Test whether the Bud14-Glc7 complex counteracts Cdc5-mediated Bfa1 inhibitory phopshorylation

Results5.2: likely so

Part two: Specific comments

– p.7 line 126 = the authors state that "deletion of mitotic exit inhibitors.... rescues growth lethality of MEN mutants" . Add references. I would like that the authors include in the dilution series experiment at least one reported strain for immediate comparison. Moreover, I slightly disagree with the conclusion of the experiment in respect to the cdc5-10 mutants. To me this mutant is not rescued by BUD14 deletion, which is ok given the central role of Cdc5 in controlling Cdc14 activity.

– p.8 I would move the concepts expressed from line 153 to 158 to the discussion

– Figure 2: (A-B) As a control I would include also a bfa1Δ bud14Δ double mutant strain in this analysis. (C) better statistical analysis

– p.10 line 192 :"Similar to bud14Δ kar9Δ (add kar9Δ)... " . Moreover, I slightly disagree with the conclusion here. The phenotype of the bud14Δ kar9Δ double mutant cells is more severe than both bud14-Φ379Α kar9Δ and glc7-ts kar9Δ associated phenotypes. This suggests an additional role of Bud14 in inhibiting mitotic exit in line with the phenotype observed in Bud14 mutants lacking the SH3 domain.

– p.11 line 215 it is Figure 3F NOT 3E

– p.12 I would move the concepts expressed from line 225 to line 234 to the discussion. As a curiosity it would be interesting to see what is the phenotype of kel1bud14 double mutant cells?

– Figure 2 - Figure supplement 1C - Please include the loading control for the western blot analysis. In this blot the 3 strains have significantly different amounts of Kin4 protein. Why is this? Overall I found it difficult to properly assess its mobility

– P.14 line 294 and 295 - Figure 5 Figure Supplement 1D not 1C

– Are the altered amounts of Bfa1, Bub2 and Tem1 at SPB affecting MEN activity? how is Dbf2 localization in cells lacking Bud14?

– I found the data presented in Figure 6 the weakest of the entire paper. (A) What is the white line between the 45 and 75 minute samples? have samples been run in different gels? If this is the case, why? Author should clearly state this. (C) I cannot fully appreciate differences in migration from the three samples. Please add a more sensitive loading control than poinceau. (D) The western Blot it is too poor to appreciate differences. To address the possibility that the Bud14-Glc7 complex antagonizes Cdc5-mediated Bfa1 phosphorylation, why not trying to assess the consequences of overexpressing Bud14 in lte1kin4 double mutant cells? Although the bulk of Bfa1 phosphorylation in lte1kin4 cells is likely due to Cdc5, why not testing directly for this introducing a cdc5 conditional allele in these cells?

– I would delete the paragraph probing for Bud14 localization. The data are not fully convincing and It does not add anything to the storyline.

[Editors’ note: further revisions were suggested prior to acceptance, as described below.]

Thank you for resubmitting your work entitled "Protein Phosphatase 1 in association with Bud14 inhibits mitotic exit in *Saccharomyces cerevisiae*" for further consideration by eLife. Your revised article has been evaluated by Anna Akhmanova (Senior Editor) and a Reviewing Editor.

The manuscript has been improved but there are some remaining issues that need to be addressed, as outlined below:

As you will see, both referees are supportive of publication, but referee 1 has suggested that it would be desirable to have the 2 hybrid results validated in a different way potentially using purified proteins or by immunoprecipitations.

I agree this is an important point for you to address.

You may already have this data, in which case, please do add it. If not, perhaps you can mention in the discussion the limitations of the conclusions based solely on the two hybrid protein-protein interaction assays.

In addition, please also address the point concerning presentation raised by referee 1.

Please find below the comments of the referees. Upon receipt of your revisions, I will decide myself and will not send it back to referees.

Essential revisions:

*Reviewer #1:*

Kocakaplan and colleagues identify Bud14 as a novel component of the Spindle Positioning Checkpoint (SPoC) in budding yeast. The SPoC is a surveillance pathway that prevents mitotic exit until the correct spindle position is achieved, hence preserving genome integrity. Importantly, this mechanism is likely to be conserved, at least functionally, in higher eukaryotes, making its characterization of high relevance. Although Bud14 was previously linked to the regulation of spindle positioning - through the modulation of Dyn1 and actin cables - this work unveils a novel connection between Bud14 and the regulation of mitotic exit. The authors propose that Bud14 in complex with the PP1 phosphatase Glc7 de-phosphorylates the checkpoint effector Bfa1, thereby fully activating it.

The authors succeeded in the characterization of Bud14 as a novel component of the checkpoint; in the identification of its role within the pathway, that is via regulation of type 1 phosphatase Glc7 and in identifying Bfa1 as a target for the Bud14-Glc7 complex. The majority of the data supporting these conclusions are solid, well controlled and sustained by the appropriate statistical analyses.

Building from genetic evidence the authors take a biochemical approach to address the molecular mechanism by which the Bud14-Glc7 complex affects Bfa1 activity. They propose that the Bud14-Glc7 complex dephosphorylates Bfa1 and this leads to activation of the checkpoint effector Bfa1-Bub2 complex. Data supporting these conclusions are somewhat convincing but can improve

I found the revised version of the manuscript significantly improved. The manuscript reads smoothly, statistics have improved and the authors succeeded in addressing the reviewers concerns.

*Reviewer #2:*

Kocakaplan et al, have shown a role for Bud14-Glc7 phosphatase in activating Spindle Positioning Checkpoint (SPoC) via Bfa1 dephosphorylation in *Saccharomyces cerevisiae*. The authors use genetic and biochemical experiments to characterize the activity of Bud14 in SPoC. Firstly, the authors show that Bud14 deletion rescues lethality of MEN-ts mutants and also causes deficient SPOC. Next, they show that the activity of Bud14 in SPoC is independent of its activity in actin regulation via Bnr1 and that Bud14-Glc7 interaction is necessary for its role in activating SPoC. They further show that Bud14-Glc7 exerts their function on SPoC via Bfa1 dephosphorylation. Their data supports a model where Bfa1 is dephosphorylated by Bud14-Glc7 complex, which may inhibit its MEN-promoting activity to activate SPoC. The authors have majorly addressed the discrepancies pointed out in the initial review, and the manuscript has greatly benefitted from the reorganization and additional experiments. The authors have done an excellent job to address all the points raised by the reviewers. The manuscript is significantly improved, and I am happy to recommend the manuscript for publication in eLife.

The authors have majorly addressed all the concerns raised by the reviewers in the initial review. The authors addressed all questions and provided additional experiments to support their claim. Overall, the manuscript is significantly improved, and I am happy to recommend the manuscript for publication in eLife.

---

## [Author Response]

[Editors’ note: the authors resubmitted a revised version of the paper for consideration. What follows is the authors’ response to the first round of review.]

Reviewer #1:In this study, the authors aim to uncover novel regulators of the Spindle position checkpoint (SPOC), and the results have implications in understanding the signaling cross-talk between mitotic exit and spindle position. The authors initially show that Bud14 deletion can rescue growth defects of mitotic exit deficient cells and based on these observations, they hypothesize that Bud14 may play a role in the activation of Spindle Position Checkpoint (SPOC). They use genetic, microscopic, and biochemical tools to test their hypothesis. The genetic and microscopic analysis presented in the manuscript demonstrates that Bud14 is involved is directly involved in SPOC activation upon spindle misalignment and is convincing despite a few minor issues. They also provide data that this effect exerted by Bud14 on SPOC may require its interaction with PP1 phosphatase Glc7; however, the involvement of Glc7 is not directly shown in all assays. Building on the genetic evidence, the biochemical analysis presented in the manuscript attempts to test whether Glc7 in association with Bud14 dephosphorylates Bfa1 and helps in its activation, but the biochemical data shown to support this claim is not sufficiently convincing and has problems in experimental design and interpretation. The data does not provide clear evidence to demonstrate that Glc7 dephosphorylates Bfa1 and lacks empirical evidence of the effect of the same on Bfa1. Overall, the study succeeds in building a preliminary case for the role of Glc7-Bud14 but lacks the mechanistic details of how they regulate bfa1 activity during normal or misaligned spindle positioning.In this manuscript Kocakaplan et al, have identifies the role of Bud14-Glc7 as a novel regulator of SPOC by controlling Bfa1 activity through dephosphorylation. This study provides a very preliminary insight into Bud14-Glc7 role in Spindle Position Checkpoint. In addition, the majority of the results presented in this manuscript need further detailed experiments and analysis to support the claims made by the authors. Some of the interpretations of data are largely not convincing (i.e., Glc7-Bud14 dependent dephosphorylation of Bfa1 in vivo and in vitro). Further mechanistic insights of Glc7-Bud14 role in regulating MEN component (Bfa1) are strongly required. Over all the initial findings by Kocakaplan et al seem interesting observations in the field of SPOC and MEN regulation.1) In Figure 1, the authors have shown that Bud14 genetically interacts with MEN components and Bud14 deletion rescues growth defects of MEN-ts mutants. Based on these observations, they hypothesize that Bud14 may be involved in mitotic exit inhibition and acting upstream of MEN pathway. The data presented support the conclusion, but the authors should mention the restrictive temperatures of MEN-ts mutants used in Figure 1C.

We now mention the restrictive temperatures of the MEN-ts mutants (page 8 lines 137-139). We also replaced the figure for this data (Figure 1C) to include *BFA1* deletion as a control for comparison of rescue of MEN-ts mutants, as suggested by Reviewer 2.

2) In Figure 2, the authors show that Bud14 deletion causes a deficiency of SPOC arrest upon spindle misalignment by comparing anaphase duration with Kar9 null cells. Bud14 is also shown to have synthetic sick interaction with Kar9. The behaviour of Bud14 is similar to the behaviour of Kin4, which is already known to be involved in SPOC activation. The data presented in the figure collectively suggest a role for Bud14 in SPOC arrest activation and supports the conclusions made by the authors. However, there are no statistics shown in Figure 2A and 2B to determine whether the differences observed are significant or not.

We now included statistics for all SPOC functionality assays shown throughout the paper including the Figure 2A and 2B. To have a simpler representation and better comparison of SPOC functionality we calculated the “SPOC deficiency index” which shows the ratio of percentage of cells with multi-nucleated phenotypes to the percentage of cells with mispositioned separated nuclei. This way of quantifying the SPOC functionality normalizes the data for fluctuations in percentage of cells with misaligned spindle among different experiments and among different samples. Therefore, it allows for better comparison and statistics.

Nevertheless, we are also including the raw data for these experiments as source data, so the percentage cell quantifications that are used for these calculations are also available. We are also including all statistical outcomes in this source data.

3) In Figure 3, the authors show that Glc7-Bud14 interaction is important for Bud14 role in SPOC and its function in SPOC is independent of its function in Bnr1-regulation. The data supports the conclusions well except for the following discrepancies:a) In Figure 3B, glc7 only rescues the lethality of lte1Δspo12Δ cells at its semi-permissive temperature (33C). This raises the question whether this condition represents a Glc7 deficient condition. The authors should address this.

We assessed *glc7-12* functionality at different temperatures based on reported phenotypes associated with Glc7 loss/reduction of function. Accordingly, phenotypes investigated were growth, HU sensitivity and accumulation of large-budded cells with single nucleus. This data is presented in Figure 4-Supplement 1. These data indicate that *glc7-12* is partially inactivated at 33 C-35 C with respect to its known functions in DNA damage response and spindle assembly checkpoint inactivation, and fully inactivated at 37 C based on growth. Our data additionally shows that this partial inactivation is also true for function of Glc7 in mitotic exit. In addition, we now show the growth phenotype of *lte1Δspo12Δglc7-12* not only at 33 C but also at 35 C in Figure 4C. Also, partial inactivation of Glc7 is now clearly stated in the text (Pages 14, lines 276290). The growth rescue and SPOC functionality of *glc7-12* is now shown in Figure 4C.

b) No statistics in Fig 3C to ascertain significance between differences.

We now included statistics for this experiment (data is now shown in Figure 4D).

c) In Figure 3D, the introduction of bud14-F379A mutant in kar9Δbud14Δ cells shows partial rescue of the multi-nucleate phenotype of kar9Δbud14Δ cells (compare black bars). This observation challenges the importance of Glc7-Bud14 interaction in activating SPOC. Also, no statistics is provided to ascertain the significance of differences.

We would like to thank reviewers for raising this point. In the previous version of the manuscript, pRS416 ADH-BUD14-F379A plasmid (pMK131) was used for SPOC functionality assays of Bud14-F379A. This plasmid, where Bud14 was expressed under ADH promoter did not allow us to detect Bud14 expression because Bud14 was untagged in the construct. To control for any difference in Bud14 expression levels, we now constructed a new plasmid that contains GFP-Bud14-F379A under the Bud14 native promoter (pHK002). This construct also let us analyze Bud14 expression levels. In the current version of the manuscript, we repeated all SPOC functionality assays using centromeric plasmids containing GFP-*BUD14* under its native promoter (pMK60, pDKY001, pDKY003 and pHK002) (Figure 3A, Figure 4B). We also analyzed the expression of all ectopic Bud14 versions used in the SPOC functionality experiments (Figure 3 - Supplement 1A). In the experiments using these plasmids, the SPOC deficiency index of *BUD14*-F379A was not statistically different than the SPOC deficiency index of cells containing the empty plasmid but was significantly different from cells expressing wild type *BUD14* (Figure 4B). Statistics for pairwise comparisons were shown in the figure as well as in the source data.

4) In Figure 4, the authors demonstrate that Bud14 interacts with Bfa1 and Bub2 is important for this interaction. The authors should also check these interactions using the already used Bud14 mutants in Figure 3 to get more insight into the nature of these interactions. In addition to Bud14, they should also check Glc7 interaction with these proteins to support the idea that Glc7-Bud14 interaction is necessary for SPOC activation.

We thank to the reviewer for this suggestion. We now show the interaction of Bfa1 with the Bud14 mutants used in the study (Figure 5C). We also analyzed interaction of Glc7 with the Bud14 mutants (Figure 3- Figure Supplement 1B) and with SPOC components (Figure 5 – Figure Supplement 1). We recapitulated the published interaction loss between Glc7 and BUD14-F379A (Figure 3- Figure Supplement 1B). Of note, Bfa1 interaction with Bud14-F379A was similar to Bfa1 interaction with Bud14 wild type in our yeast two hybrid assays (Figure 5C), whereas Glc7 interaction with Bud14-F379A was completely diminished. This data shows that Bud14-F379A is defective in Glc7 binding but not in Bfa1 binding and supports the use of Bud14-F379A to assess contribution of Glc7-Bud14 to mitotic exit, SPOC and Bfa1 hyperphosphorylation.

We also recapitulated the published interaction loss between Glc7 and Bud14ΔSH3 mutant (Figure 3- Figure Supplement 1B). We further found that interaction of Bfa1 with Bud14ΔSH3 mutant was reduced when compared to Bfa1 interaction with the wild type Bud14 (Figure 5C). With this data we can now explain the SPOC functionality loss observed in the Bud14ΔSH3 mutant.

Curiously, we also observed that the interaction of Glc7 with Bud14-5A was enhanced compared to the Glc7-Bud14 wild type interaction (Figure 3- Figure Supplement 1B). However, interaction of Bfa1-Bud14 and Bfa1Bud14-5A was similar in our yeast two hybrid assays (Figure 5C). This data now can explain the significantly increased SPOC proficiency (decreased SPOC deficiency index) of the Bud14-5A mutant (Figure 3A).

5) In Figure 5, the authors show that more Bfa1 associates with SPBs, especially dSPB in Bud14Δ cells. They also show that unlike Kin4 deletion, Bud14 depletion does not change in the asymmetric localization of Bfa1 between the two SPBs but merely causes an increase in Bfa1 accumulation at SPBs irrespective of spindle misalignment. These observations suggest a role for Bud14 in controlling Bfa1 sequestration at the SPB and further support the idea that Kin4 and Bud14 work independently in different pathways in SPOC.

We now mentioned in the text that this data is in support of the idea that Kin4 and Bud14 work independently in different pathways in SPOC (Page 11 lines 224-228, Page 17 lines 369-372).

6) In Figure 6, the authors show that Bfa1 may be dephosphorylated by Bud14 as there is the accumulation of hyper-phosphorylated Bfa1 in lte1Δbud14Δ and lte1Δkin4Δ cells. The authors argue that like Kin4, Bud14 activity may be counteracting the effect of Cdc5 kinase-mediated phosphorylation of Bfa1. However, the data has the following discrepancies:7) In Figure 6C, the effect of the phosphatase inhibitor Okadaic acid is not striking even at high concentration (lane 3). Since the effect is so small, I suggest the authors to quantify the effect for better visualization of the reader.

We agree that the effect of the inhibitor is minor in this experimental setup. This is most likely due to the already hyperphosphorylated nature of Bfa1 in the anaphase cell extracts. With the suggestion from the Reviewer 2, we now replaced this experiment with another experimental set up, which allowed us better visualization of the differences in Bfa1 migration (page 18-19, lines 389-398, Figures 7D, 7E, 7F). We now show the phosphorylation status of Bfa1 from anaphase arrested cells (via usage of analog sensitive allele of Cdc15-as) with and without overexpression of *BUD14* (Figure 7D-E) and *bud14-F379A*, which cannot bind Glc7 (Figure 7E-F). Our results show that overexpression of *BUD14* but not the *Bud14-F379A* causes a downshift in Bfa1 phosphorylation. This finding indicates a Bud14-Glc7 dependent Bfa1 dephosphorylation.

8) Based on Figure 6C, the authors make the conclusion that Glc7 promotes dephosphorylation of Bfa1 (lines 342-343 in text), but Okadaic acid is not a specific inhibitor for Glc7. It can also inhibit Cdc55-PP2A and other phosphatases (i.e., Cdc14), which is known to dephosphorylate Bfa1 so this conclusion cannot be made until the contribution of Cdc55 can be separated from the contribution of Glc7.

The new experiment (Figure 7D-E) did not require the use of Okadaic acid which also avoids any concern of unspecific inhibition of PP2A along with PP1. Nevertheless, we were curious about the contribution of Cdc55 on Bfa1 dephosphorylation during anaphase. Although we did not show it in manuscript, we monitored the contribution of Cdc55 on Bfa1 hypophosphorylation in *lte1Δ* cells in the same experiment shown in Figure 7A-B. Unlike *bud14Δ*, *cdc55Δ* did not cause hyperphosphorylation of Bfa1 in *lte1Δ* cells during anaphase. This was consisted with the previous reports that showed PP2A-CDC55 promotes Bfa1 dephosphorylation in metaphase, and at anaphase onset Cdc55PP2A activity is downregulated in a Separase-dependent manner to allow Cdc5 phosphorylation of Bfa1 and thus MEN activation (Baro *et al*, 2013; Queralt *et al*, 2006).

9) In Figure 6D, the bands in Lane 1,2,3 show very little presence of Bfa1-HA (measured using anti-HA antibody) as compared to Lanes 4,5,6. Considering that equal amounts of Bfa1-HA were used for the in vitro phosphatase assay, the levels of Bfa1-HA in all lanes should be approximately equal. This can potentially confound analysis of the assay. The authors should also check Bfa1 dephosphorylation in the presence of both Bud14 and Glc7 as that represents the in vivo condition. The experiment should be replicated to prove the reproducibility of the observed result.

We apologize for the loading problem in this figure. We replicated the experiment and replaced the figure with a new one (now shown as Figure 7G and Figure 7- Figure Supplement 2B). We also included a quantification of the observed changes in Bfa1 phosphorylation by presenting the ratio of intensity of the slow migrating Bfa1 forms to the intensity of fast migrating Bfa1 forms. For both assays that used Glc7TAP (Figure 7G) and Bud14-TAP (Figure 7-Figure supplement 2B) we performed 4 independent experiments. These experiments show that addition of Glc7-TAP or Bud14-TAP causes a downshift in Bfa1, which was greatly prevented by inhibitor addition. Importantly, the downshift of Bfa1 upon Glc7-Bud14 addition was consistent in all experiments and the differences were statistically significant.

It is also worth stating that due to inherent variabilities in these phosphatase reactions, the variation among independent experiments is relatively high. The variation comes from several factors:

1. Glc7 and Bud14 are pulled down on IgG beads and used freshly in thereaction in each experiment.

2. For each experiment Bfa1-3HA was purified from anaphase arrestedyeast cells freshly, as the hyperphosphorylated Bfa1-3HA diminished upon storage.

3. Okadaic acid was prepared freshly for each experiment, as we realizedlong term storage was problematic in our hands.

4. Separation of Bfa1 phospho-forms in Western blot is inherentlyvariable, depending on variation in the home-made gels, buffers, blotting etc.

10) In Figure 6D, Bud14 addition has no visible effect on the phosphorylation status of Bfa1, but the authors come to a totally opposite conclusion in Lines 349-352 in the text.

We are sorry for this confusion. As explained above in detail (see response to point 9), we now show another blot for Bud14 addition with the control lane loaded just next to Bud14 (Figure 7-Figure supplement 2B). We also included a quantification of this experiment. The data comes from 4 independent experiments. The pattern was consistent in all experiments.

11) In Figure 7, the authors show that the percentage of cells with Bud14 showing bud cortex localization decreases upon spindle misalignment, but this could just be a consequence of prolonged mitotic arrest due to SPOC. The data presented in the figure does not contribute much to the main story of the paper. As an addition, the authors should also check co-localization of Bud14, Glc7 and Bfa1 during the cell cycle in cells with normal and misaligned spindles to test their hypothesis that Glc7-Bud14 complex dephosphorylates Bfa1 to activate SPOC.In summary, the manuscript needs major revision with more emphasis on uncovering the molecular mechanism of how Glc7-Bud14 action on Bfa1 helps in SPOC.

We agree that this data does not contribute much to the main story of the paper as the localization change is likely a consequence of prolonged mitotic arrest. We removed this figure from the current version of the manuscript as suggested by both reviewers. However, it is worth mentioning that we performed more localization experiments, which we would like to share with the reviewers only. We did check Glc7 localization and observed that Glc7 localizes to the poles of the spindles during anaphase regardless of the spindle position. Glc7 was already shown to be an interaction partner of kinetochore proteins and to be localizing at the poles. Because kinetochores of the budding yeast cluster underneath the SPBs, we don’t know whether this localization is at the SPBs or at the kinetochore clusters. To discriminate this, we used a temperature sensitive mutant (*ndc10-1*) in which chromosomes/kinetochores remain only at one pole during anaphase. However, as we shift the temperature from 30 to 37 C for inactivation of *ndc10-1*, Glc7 pole localization become completely undetectable in both wild type and *ndc10-1* cells. Glc7 localization at poles is normally very faint and we think that the raise in the temperature may cause an increase in the background noise which may interfere with the signal detection even in wild type cells. Therefore, at the moment, we are unable to address whether Glc7 is at SPBs or kinetochores during spindle misalignment. Nevertheless, we are not showing any of this data and as suggested by both reviewers we removed Bud14 localization data from the manuscript.

Reviewer #2:[…]Although I found the topic interesting and most of the data solid, I found the manuscript very difficult to read. My major criticism of the manuscript concerns the logic of the storyline. In particular it was not always clear to me the rationale by which experiments were performed and interpreted and consequently I found confusing the sequence in which the data were presented. I believe that the manuscript will benefit from a substantial re-writing and re-organization of the data. For the above-mentioned reason I organized my review into two distinct sections: the first aimed at suggesting an alternative way of presenting the data, the second focused on addressing few experimental concerns, whose elucidation could contribute in strengthening the authors conclusions.Part one:Following the identification of Bud14 as a novel mitotic exit inhibitor, the authors move to investigate its potential role in the SPoC. I would re-organize the manuscript starting from here:Q1: Is Bud14 a component of the SPoC?Rationale1: Inactivation of SPoC in cells with mis-aligned spindles results in formation of multinucleate and anucleate cells.Experimental design1: assess the consequences of deleting BUD14 in cells lacking Kar9 or Dyn1 - both presenting mis-aligned spindles (I would introduce Dyn1 with the same logic used for Kar9) - looking at multinucleate cells as outputResults1: In both cases the percentage of multinucleate cells increases indicating that bud14Δ cells are defective in SPoCRationale1.2: To further investigate the role of Bud14 in SPoC assess the anaphase residence timingResults1.2: Bud14 is a bona fide SPoC componentQ2: Is Bud14 working together or parallel to Kin4?Rationale2: Epistasis experiments: kin4 bud14 double mutant and GAL-KIN4 bud14Results2: They work in parallelQ3: Which function of Bud14 is required for its SpoC activity?Rationale3: Bud14 is reported to act as an inhibitor of formin and a regulatory subunit of PP1 Glc7. Assess whether one of these functions is required for SPoC activityExperimental design3.1: Formin/actin experimentResults3.1: Bud14 does not contribute to the SpoC via formin or actin regulationExperimental design3.2: Test for Glc7 functionResults3.1: Bud14 contributes to SPoC activity at least in part through its association with Glc7. However the obs that lack of the SH3 domain leads to a more severe defect calls for an additional mechanism, likely involving Bud14 cortex localizationQ4: What is Bud14-Glc7 target within the SPoC?Rationale4: the obs that: i)Kin4 targets Bfa1; ii)the multinucleate defect of kin4bud14 double mutants in kar9delta cells resembles the one of kar9bfa1 double mutants identify Bfa1 as a putative candidateExperimental design4: 2-hybrid experiment and phosphorylation/de-phosphorylation experimentsResults4: Bfa1 is a target of the Bud14-Glc7 complexQ5: How does Bud14-Glc7 affect Bfa1 activity?Rationale5.1: Bfa1 regulated by changes in subcellular localizationExperimental design5.1: Test Bfa1 localization in cells lacking Bud14Results5.1: does not seem to affect localization but rather Bfa1 amountQ6: Does Bud14-Glc7 counteract Cdc5-mediated Bfa1 inhibitory phopshorylation?Rationale5.2: Bfa1 is inhibited by Cdc5Experimental design5.2: Test whether the Bud14-Glc7 complex counteracts Cdc5-mediated Bfa1 inhibitory phopshorylationResults5.2: likely so

We are grateful to the reviewer for the suggestion. It was a great contribution to the manuscript. We re-organized the manuscript in light of her suggestions. We think that the story line is more logical in this version of the manuscript, and it is now easier to grasp the rationale behind experiments.

Part two: Specific comments– p.7 line 126 = the authors state that "deletion of mitotic exit inhibitors.... rescues growth lethality of MEN mutants" . Add references. I would like that the authors include in the dilution series experiment at least one reported strain for immediate comparison. Moreover, I slightly disagree with the conclusion of the experiment in respect to the cdc5-10 mutants. To me this mutant is not rescued by BUD14 deletion, which is ok given the central role of Cdc5 in controlling Cdc14 activity.

We now also added *bfa1Δ* as a reported strain for immediate comparison (Figure 1C). We also added references (page 7, line 135). After comparing with the growth of *bfa1Δ* strain, it became more obvious that *BUD14* deletion did not rescue the *cdc5-10*. We clearly stated this in the manuscript (page 8, lines 140-142). This result could be because of the central role of Cdc5 in controlling Cdc14 activity as also stated by the reviewer. Additionally, it could also be because of lack of Bfa1 hyperphosphorylation in the *cdc5-10* mutant. Of note, *cdc5-10* lethality can be rescued by *BFA1* deletion at 37 C, indicating that this lethality is partly due to hyperactive Bfa1 (due to lack of hyperphosphorylation by Cdc5). Considering the function of Glc7-Bud14 in dephosphorylation of this hyperphosphorylated Bfa1, Bud14-Glc7 function in MEN inhibition is expected to become dispensable in *cdc5-10*, in which Bfa1 hyperphosphorylation is absent.

– p.8 I would move the concepts expressed from line 153 to 158 to the discussion

In the new version of the manuscript, we mentioned these concepts only in the discussion (Page 21, lines 438-441).

– Figure 2: (A-B) As a control I would include also a bfa1Δ bud14Δ double mutant strain in this analysis. (C) better statistical analysis

We added this control in the new version of the manuscript (Figure 2A, page 10, lines 192-194) and we performed one-way Anova as statistical analysis of the data shown in Figure 2C. All the statistics of the pairwise comparisons were shown in the source data for Figure 2C, whereas only comparisons of normal and misaligned spindles were indicated on the figure to avoid cluttering of the graph.

In addition, as was raised by reviewer #1 we now included statistics for all population-based SPOC functionality assays shown throughout the paper including the Figure 2A and 2B. To have a simpler representation and better comparison of SPOC functionality we calculated the “SPOC deficiency index” which shows the ratio of percentage of cells with multinucleated phenotypes to the percentage of cells with mispositioned separated nuclei. This way of quantifying the SPOC functionality normalizes the data for fluctuations in percentage of cells with misaligned spindle among different experiments and among different samples. Therefore, it allows for better comparison and statistics. Nevertheless, we are also including the raw data for these experiments as source data, so the percentage cell quantifications that are used for these calculations are also available. We are also including all statistical outcomes in this source data.

– p.10 line 192 :"Similar to bud14Δ kar9Δ (add kar9Δ)... " . Moreover, I slightly disagree with the conclusion here. The phenotype of the bud14Δ kar9Δ double mutant cells is more severe than both bud14-Φ379Α kar9Δ and glc7-ts kar9Δ associated phenotypes. This suggests an additional role of Bud14 in inhibiting mitotic exit in line with the phenotype observed in Bud14 mutants lacking the SH3 domain.

Thank you for pointing this out. The same concern for BUD14-F379A was also raised by the reviewer #1 (point 3c). Here is our response:

In the previous version of the manuscript, pRS416 ADH-BUD14-F379A plasmid (pMK131) was used for SPOC functionality assays of Bud14F379A. This plasmid, where Bud14 was expressed under ADH promoter did not allow us to detect Bud14 expression because Bud14 was untagged in the construct. To control for any difference in Bud14 expression levels, we now constructed a new plasmid that contains GFP-Bud14-F379A under the Bud14 native promoter (pHK002). This construct also let us analyze Bud14 expression levels. In the current version of the manuscript, we repeated all SPOC functionality assays using centromeric plasmids containing GFP-*BUD14* under its native promoter (pMK60, pDKY001, pDKY003 and pHK002) (Figure 3A, Figure 4B). We also analyzed the expression of all ectopic Bud14 versions used in the SPOC functionality experiments (Figure 3 - Supplement 1A). In the experiments using these plasmids, the SPOC deficiency index of *BUD14*-F379A was not statistically different than the SPOC deficiency index of cells containing the empty plasmid but was significantly different from cells expressing wild type *BUD14* (Figure 4B). Statistics for pairwise comparisons were shown in the figure as well as in the source data.

Furthermore, we now have an explanation for the phenotype of Bud14 lacking the SH3 domain. Based on yeast two hybrid analysis, we showed that interaction was abolished between Glc7 and Bud14ΔSH3 mutant (Figure 3- Figure Supplement 1B). This interaction loss was also published before (Knaus *et al*, 2005). Through yeast two hybrid assays, we further found that interaction of Bfa1 with Bud14ΔSH3 mutant was also reduced (Figure 5C). Thus, the loss of SPOC function in Bud14ΔSH3 mutant can be attributed to loss of its interaction with Glc7 and Bfa1.

– p.11 line 215 it is Figure 3F NOT 3E

This is changed in the current version of the manuscript (page 12, line 238). The corresponding figure is shown as Figure 3A.

– p.12 I would move the concepts expressed from line 225 to line 234 to the discussion. As a curiosity it would be interesting to see what is the phenotype of kel1bud14 double mutant cells?

We were also curious to see the phenotype of *kel1Δbud14Δ* double mutants. We analyzed the phenotype with respect to the ability to rescue *lte1Δspo12Δ* lethality and to cause SPOC deficiency. The SPOC deficiency of the double mutant was similar to the SPOC deficiency of the *bud14Δ*. Double mutants had a growth rescue phenotype more than the *kel1Δ* single mutant. The growth rescue of the double mutant was comparable to the rescue by *bud14Δ*, but the colony sizes were slightly reduced. This reduction was not observed on SC plates.

This data is in line with that Bud14 has more profound role in inhibition of MEN, and also suggests that the cellular roles of Kel1 and Bud14 may be more complex than we anticipated. We decided not to show this data in the manuscript to keep it simple and focused. Conclusions made in the manuscript are not contradicting with this data.

As we want to show the growth rescue differences between *bud14Δ*, *kel1Δ* and *kel2Δ* (in the *lte1Δ* and *lte1Δspo12Δ* backgrounds), we decided to keep this part of the manuscript in the results part (page 13, lines 255264). It can however be moved to the discussion as “data not shown” if necessary.

– Figure 2 - Figure supplement 1C - Please include the loading control for the western blot analysis. In this blot the 3 strains have significantly different amounts of Kin4 protein. Why is this? Overall I found it difficult to properly assess its mobility

We apologize for this blot. We now show another blot which also has a loading control (Figure 2 – Figure supplement 1D). Kin4 is hyperphosphorylated in *rts1Δ* cells, while this is not the case for wild type and *bud14Δ* cells.

– P.14 line 294 and 295 - Figure 5 Figure Supplement 1D not 1C

We are sorry for this mistake. This is changed in the current version of the manuscript (page 17, lines 348-349). The corresponding figure is shown as Figure 6 – Figure Supplement 1D in the current version of the manuscript.

– Are the altered amounts of Bfa1, Bub2 and Tem1 at SPB affecting MEN activity? how is Dbf2 localization in cells lacking Bud14?

Thank you for this question. To understand whether the increased Bfa1Bub2 and Tem1 recruitment to the dSPBs observed in *bud14Δ* cells may be affecting MEN activity, i.e may be causing premature activation of the MEN, we analyzed Mob1 SPB localization in these cells. We chose Mob1 over Dbf2 due to its brighter fluorescence signal. Mob1-GFP did not prematurely localize to SPBs in *bud14Δ* cells (Figure 6 – Figure Supplement 2A-C). The levels of Mob1-GFP at the SPBs of *bud14Δ* cells were not altered in anaphase either (Figure 6- Figure Supplement 2D-E). Thus, we conclude that despite the elevated levels of Bfa1-Bub2 and Tem1 at the dSPB, MEN activity is not enhanced by *BUD14* deletion in cells with normally aligned spindles. This experiment results are presented in the text (page 16, lines 336-345) and in Figure 6- Figure Supplement 2.

– I found the data presented in Figure 6 the weakest of the entire paper. (A) What is the white line between the 45 and 75 minute samples? have samples been run in different gels? If this is the case, why? Author should clearly state this.

We apologize for the confusion. The samples were run in the same gel, but due to loss of the 60 min *bud14Δ* sample during protein extraction, we removed the lanes for 60min for all three samples from the image and added the white line instead, to clearly show that a lane was removed. To avoid confusion, we now show another blot from another experiment (Figure 7A-B). *lte1Δkin4Δ* and *lte1Δ* time-course series from the previous figure is now shown in another figure without the white line (Figure 7 – Figure Supplement 1).

C) I cannot fully appreciate differences in migration from the three samples. Please add a more sensitive loading control than poinceau.

We agree that the effect of the inhibitor is minor in this experimental setup. This is most likely due to the already hyperphosphorylated nature of Bfa1 in the anaphase cell extracts. Also considering the concerns raised by the Reviewer #1 (points # 7 and 8) with respect to usage of okadaic acid, this experiment is now replaced with Bud14 overexpression experiments (Figure 7D-E).

D) The western Blot it is too poor to appreciate differences.

(See also Reviewer #1 concern 9 and10). We apologize for the blot problem in this figure. We now repeated the experiments and replaced the figure with new ones (now shown as Figure 7G and Figure 7- Figure Supplement 2B). We also included a quantification of the observed changes in Bfa1 phosphorylation by presenting the ratio of intensity of the slow migrating Bfa1 forms to the intensity of fast migrating Bfa1 forms. Importantly, the downshift of Bfa1 upon Glc7 addition was consistent in all experiments and the differences were statistically significant (now shown as Figure 7G and Figure 7- Figure Supplement 2B). We believe that with these quantifications and data representation, the differences in Bfa1 mobility among samples are more noticeable.

It is however worth stating that due to inherent variabilities in these phosphatase reactions, the deviation among independent experiments is relatively high. The variation comes from several factors: 1. Glc7- and Bud14 are pulled down on IgG beads and used freshly in the reaction in each experiment. 2. For each experiment Bfa1-3HA was purified from anaphase arrested yeast cells freshly, as the hyperphosphorylated Bfa13HA diminished upon storage. 3. Okadaic acid was prepared freshly for each experiment, as we realized long term storage was problematic in our hands. 4. Separation of Bfa1 phospho-forms in Western blot is inherently variable, depending on variation in the home-made gels, buffers, blotting etc…

To address the possibility that the Bud14-Glc7 complex antagonizes Cdc5-mediated Bfa1 phosphorylation, why not trying to assess the consequences of overexpressing Bud14 in lte1kin4 double mutant cells?

We would like to thank to the reviewer for suggesting this very important experiment. We analyzed the effect of *BUD14* overexpression on Bfa1 phosphorylation (page 18-19, lines 389-398, Figures 7D, 7E, 7F). As Bud14 overexpression causes cells to arrest in a pre-anaphase state, we first arrested *cdc15-as* bearing cells in anaphase through 1NM-PP1 treatment and then induced Gal1-*BUD14* overexpression by addition of Galactose into the growth medium. Overexpression of *BUD14* (Figure 7D) caused a downshift in Bfa1 migration in anaphase arrested cells. We also analyzed the effect of *bud14-F379A* overexpression in the same experimental setup (Figure 7E). *bud14-F379A* overexpression did not cause a downshift in Bfa1 phosphorylation, while *BUD14* overexpression did (Figure 7E). The relative ratios of slow migrating and fast migrating forms of Bfa1 in this experiment were also presented (Figure 7F). With this data, now, we can clearly show the effect of Glc7-Bud14 on Bfa1 phosphorylation.

Although the bulk of Bfa1 phosphorylation in lte1kin4 cells is likely due to Cdc5, why not testing directly for this introducing a cdc5 conditional allele in these cells?

We now show that Bfa1 phosphorylation in *bud14Δlte1Δ* cells is Cdc5 dependent, using the *cdc5-10* conditional allele (Figure 7C and page 18, lines 386-388).

– I would delete the paragraph probing for Bud14 localization. The data are not fully convincing and it does not add anything to the storyline.

As suggested by both reviewers, we now removed this data. Please also see the response to the reviewer #1, major point 11, as we presented therein the results of additional experiments regarding Bud14-Glc7 localization.